# VH-replacement shapes the antibody repertoire by removing the genes of non-functional heavy-chains

Harry N White [1]✉, Peter Chovanec[2], Laura Biggins [1], Elise C French[1], Georgia Bullen[1], Simon Andrews [1] & Anne E Corcoran[1]

## Abstract

**The diversity of antibodies underpins robust immune responses. During the formation of the antibody repertoire in early bone marrow B-cells, random antibody heavy-chain proteins are generated from recombined VH, DH, and JH gene segments. Many are non-functional and are negatively selected. To understand this process in normal mice, we have undertaken an in-depth analysis of heavy-chain selection at this pre-B cell transition. We find independent selection acting on three regions of the complementarity-determining region 3 (CDR3) antigen-binding site, with particularly heavy counter-selection against certain productive VH/JH combinations. This led us to hypothesise that VH-replacement, where the VH gene segment in an existing VDJ combination is replaced, targets productive VDJ rearrangements that code for non-functional heavy chains. We detect VH-replacement recombination products that closely follow the pattern of selection of functional and non-functional VDJ rearrangements. This reveals a physiological role for VH-replacement in the developmental release of B-cells that are stalled by non-functional heavy-chains. This leads to re-modelling of the restricted early VDJ repertoire toward the use of other VH gene segments throughout the *IgH* locus.**

**Keywords** B-cell Development; VH-replacement; VDJ-recombination; Antibody Repertoire
**Subject Categories** DNA Replication, Recombination & Repair; Immunology

## Introduction

The enormous diversity of antibodies is generated during B-cell development in the bone marrow. This diversity is largely facilitated by the semi-random recombination of antibody heavy-chain V, D and J gene segments. VDJ recombination is catalysed by the RAG recombinase (Schatz et al, 1989) with D–J joining occurring first, usually on both alleles, in early pro B cells (Alt et al, 1984). Through imprecise joining, the junction regions between D–J and V–D are further diversified (Weigert et al, 1980).

Combined with the D-segments, which can be read in any translational reading frame (RF), they form the highly diverse heavy-chain CDR3(H) regions (Fig. 1A).

As a result of the imprecision of recombination, many VDJ cannot be translated into antibody heavy-chain proteins, and of those that can, many code for heavy chains that cannot function in antibodies. After the initial VDJ recombination, a developmentally regulated sequence of events selects the heavy chains suitable for use in antibodies.

The D-segment RF has a profound impact on the encoded amino acids in the CDR3 (Fig. 1A). A first selection occurs against recombinants using RF2 and RF3. In-frame with most D-regions in RF2 in mouse B cells is an ATG start codon. If the initial D–J recombination is in RF2, this leads to expression of a truncated 'Dμ' protein and developmental arrest (Tornberg et al, 1998). Many D-regions contain stop codons in RF3 (Fig. 1A), resulting in counter-selection. The precise impacts of RF2 and RF3 counter-selection are unknown. Human B-cells use other processes to select RFs as Dμ-selection doesn't occur (Minegishi and Conley, 2001).

In pro-B cells, VDJ rearrangements that are in-frame and without stop codons, 'productive', are expressed as heavy-chain—'μ' protein. Co-expressed with μ is the SLC, formed from two components, VpreB and λ5, which may pair with μ to form the pre-B cell receptor (pre-BCR). Successful pre-BCR signalling from a 'functional' pairing heavy chain, results in suppression of SLC expression (Grawunder et al, 1995) and suppression of further VDJ recombination at the second heavy-chain (*Igh*) allele (allelic exclusion). This is followed by several cell divisions of large pre-B cells, finishing the 'pre-B cell transition' and producing small pre-B cells ready for light-chain recombination (Martensson et al, 2010).

Importantly, as many as half of new μ-chains cannot pair with the SLC, and pairing efficiency differs between VH families (ten Boekel et al, 1997), although sample numbers in this study were low.

A pre-BCR structure shows that particular amino acid residues of the SLC make extensive contact with the highly variable junction residues that bracket the CDR3H, forming a 'sensing site' responsive to sequence variation (Bankovich et al, 2007). Thus, variable stability of μ-chain/SLC pairing, which impacts the longevity of the pre-BCR (Lassoued et al, 1996), drives differential VDJ selection over the pre-B transition (Malynn et al, 1990).

Poor μ-chain pairing with the SLC could be V-segment (VH)-intrinsic, depend on the CDR3 sequence, or be a combination of

[1]The Babraham Institute, Cambridge CB22 3AT, UK. [2]David Geffen School of Medicine, UCLA, Los Angeles, CA 90095, USA. ✉E-mail: harry.white@babraham.ac.uk

both. Specific instances of CDR3 sequence selection over the pre-B transition have been reported. Those few VH5-2/81X heavy chains that survive the pre-B transition often use Histidine at position 99 in the CDR3H (Martin et al, 2003). Similarly, selection for Tyrosine residues in VH position 101 in VH5/7183 genes occurs (Khass et al, 2016). Neither example, however, has resolved the question of whether these CDR3 sequence selections vary with different VH.

An additional process that may contribute to repertoire alteration at the pre-B transition is VH-replacement (VHR). Here, a pre-existing VDJ is invaded by an upstream VH on the same allele, that replaces the original VH creating a new VDJ (Koralov et al, 2006; Usuda et al, 1992). Most VH have a 'cryptic' 7-mer recombination-signal-sequence (cRSS) close to their 3′ end that could facilitate this. The impact of VHR on physiological VDJ repertoire organisation is unknown. It was thought to enable heavy-chain receptor-editing (Chen et al, 1995), but this was disproven (Sun et al, 2015). VHR occurs in pro-B cells and is Rag-dependent (Cowell et al, 2003; Davila et al, 2007). In a mouse carrying a monoclonal functional VDJ, subsequent VHR can contribute up to 20% of the peripheral antibody repertoire (Kumar et al, 2015).

We have conducted an in-depth analysis of antibody heavy-chain selection over the pre-B transition using quantitative VDJ repertoire analysis. We find strong selection against particular VH-family/JH combinations, selection of key CDR3 amino acid residues in a VH-dependent and independent manner, and efficient RF2 counter-selection after Dμ-selection. Further investigation of the VDJ counter-selection in pro-B cells revealed VH-replacement associated with particular VDJ, many of which are known not to pair with the SLC.

## Results

### Progenitor B-cell definition and immunoglobulin heavy-chain repertoire analysis

We undertook an in-depth analysis of VDJ selection over the pre-B transition, focusing on the VH repertoire of pro-B cells (Basel 'pre-BI') and small pre-B cells (Basel 'small pre-BII'), sorted according to the scheme in Fig. EV1A. This adapts the use of markers defined by Hardy (Hardy et al, 1991) and the Basel group (Rolink and Melchers, 1996). We find this approach more stringently defines pro-B cells. In our hands the conventional FACS sorted CD43+/IgM-/c-kit+ pro-B cell phenotype includes a subset of cells with a large pre-B cell phenotype (Fig. EV1B). B-cell genomic DNA was then subject to quantitative immunoglobulin VDJ repertoire analysis using VDJseq (Bolland et al, 2016; Chovanec et al, 2018). This next-generation sequencing technique captures J-segment-containing sequences and detects both DJ and VDJ recombinants, allowing measurement of the overall frequency of VDJ recombinants in populations. The VDJseq approach is highly reproducible, showing strong correlation for biological replicate samples (Fig. EV1C). Replicate samples, thus, were merged after the VDJseq pipeline. VDJseq has a sampling rate of around 5–10% (Chovanec et al, 2018). Use of bone marrow (BM) cell pools from a minimum of four mice, and the merging of data from up to three biological replicate pools (Appendix Table S1) allowed us to accurately determine VDJ frequencies and CDR3 properties even from subsets of the rare pro-B cell population.

We analysed the 89 most frequently used VH in VDJ recombinants from pro-B cells, that form 98% of the repertoire (Appendix Table S2). The sequence metadata are in Appendix Table S1. μMT mice were also analysed; the μMT mouse has a deletion in the IgM transmembrane domain abrogating signalling and causing a developmental block at the pro-B cell stage (Kitamura et al, 1991). Due to minor differences we detect between the C57BL/6 (wild-type, WT) and μMT, described in 'Methods', we are not directly comparing VH repertoire frequencies between these strains, but using μMT mice to analyse other fundamentals of VDJ recombination.

In this report, we use the expression 'productive' to describe a VDJ rearrangement that can be translated into μ-chain protein, and 'functional' to describe a productive VDJ rearrangement encoding a μ-chain that can successfully pair with the SLC and drive pre-BCR-mediated proliferation. We also adopt the convention that the conserved Alanine residue is the beginning of the CDR3H and is located at VDJ position 97.

### Less than a quarter of heavy-chain VDJ rearrangements are productive

The heavy chain in μMT mouse pro-B cells cannot signal, so these cells should provide an unbiased measure of the frequency of generation of productive VDJ rearrangements. We found that only 23% of VDJ recombinants are productive in μMT mouse pro B-cells (Fig. 1B; Appendix Table S1).

### D reading-frame selection halves the frequency of VDJ with RF2 and RF3

Dμ-selection, which does not occur in μMT mice (Gu et al, 1991), halves the relative frequency of RF2 VDJs (Fig. 1C,D). Stop codons more than halve productive rearrangements in RF3 (Fig. 1C). No D-regions have a stop codon in RF2, in the overwhelmingly favoured forward orientation. Thus, the 10% of RF2 in-frame VDJ with a stop codon indicates the frequency of stop codons are formed de novo during VDJ recombination (Fig. 1E).

### VH selection over the pre-B transition

Figure 1F shows there is little or no VH-specific change in the levels of productive VDJ compared to non-productive VDJ in μMT pro-B cells.

Our analysis of VH selection in wild-type mice over the pre-B transition detects selection occurring in two stages. Firstly, in contrast to μMT pro-B cells, wild-type pro-B cells show VH-specific selection of productive VDJs (Figs. 1G and EV2). This selection is, therefore, dependent on μ-chain signalling but, interestingly, has occurred prior to any pre-BCR-mediated proliferation.

Secondly, there is further VH-selection after completion of the pre-B transition (Figs. 1H and EV3), likely driven by differential SLC-pairing and pre-BCR-mediated proliferation.

Figure 2A shows the overall VH selection over the pre-B transition; from pro-B non-productive VDJ to small pre-B productive VDJ, plotted by Igh locus position. Figure 2B shows the classical names of the VH families for reference. The gene most proximal to the D-segments in the locus, VH5-2 (81X), shows a

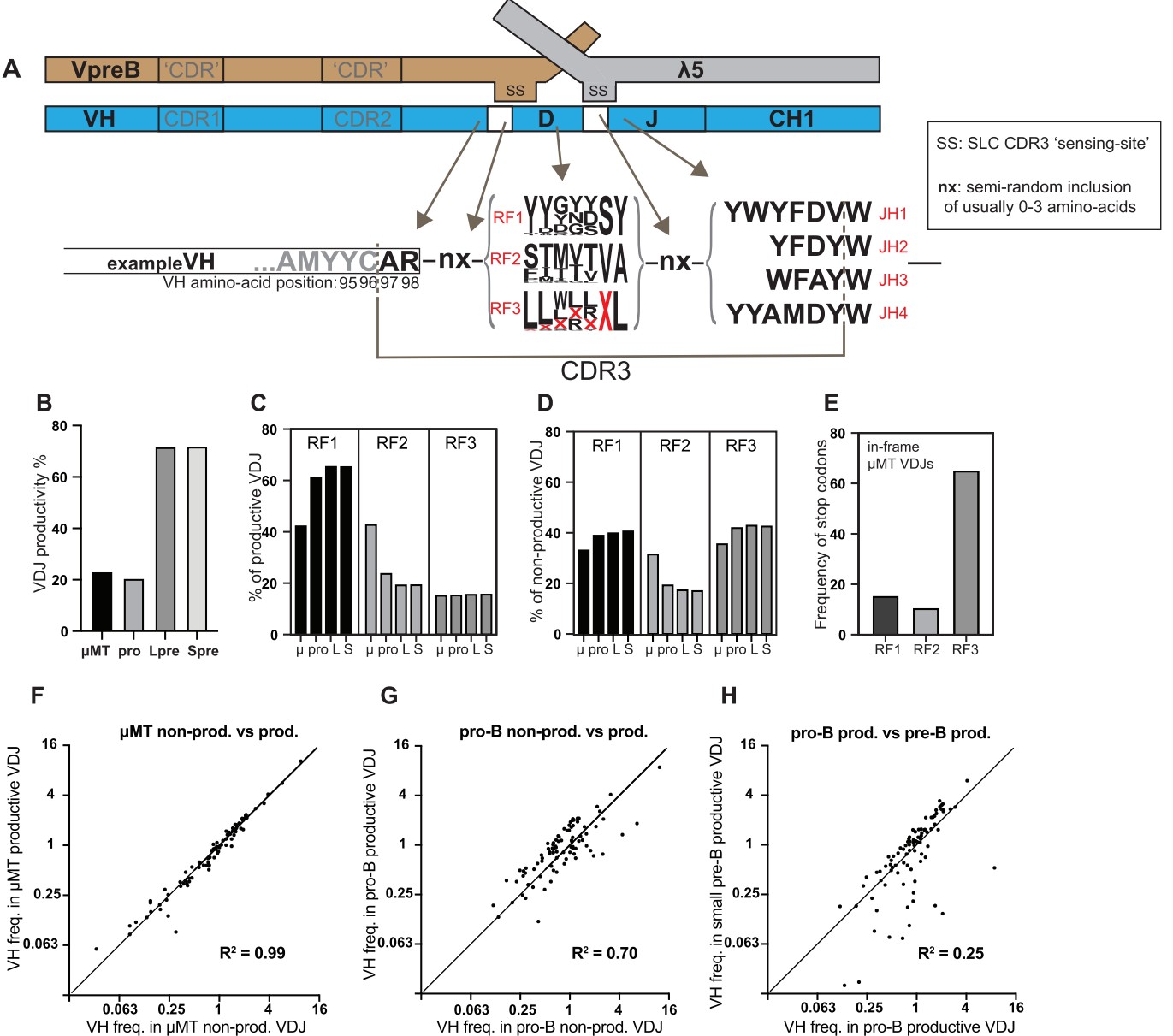

**Figure 1. CDR3 sequence structure and overall VDJ productivity, reading frames and stop codons.**

(A) Amino acid sequence structure of the heavy-chain CDR3 showing the contribution of different gene segments and random elements to the CDR3 sequence. We use the convention that the invariant Cysteine at the 3′ end of the VH framework-3 region (FR3) is residue 96, and the CDR3 begins at residue 97 and ends before the conserved Tryptophan (W) residue in JH. The logo plot of frequencies of amino acid use in the different D reading frames, RF1-3, was calculated using D-region sequences factored by their frequency of use in μMT pro-B cells. The red X represents a stop codon. The regions nX usually result in the net inclusion of 0 to 3 amino acids. This outcome is the result of the inclusion of palindromic and non-templated nucleotides by Artemis hairpin-cleavage and TdT activity, respectively, and also exonuclease removal of nucleotides, all during VDJ recombination. Thus, the terminal residues of the VH, D and JH can also be altered. (B) Frequency of productive VDJ formation in the four cell types analysed: μMT, μMT pro-B cells; pro, pro-B cells; Lpre, large pre-B cells; Spre, small pre-B cells. (C) Relative frequency of different D-reading frames in productive VDJ from the four cell-types analysed: μ, μMT pro-B cells; pro, pro-B cells; L, large pre-B cells; S, small pre-B cells. (D) Relative frequency of different D-reading frames in non-productive VDJ from the four cell-types analysed: μ, μMT pro-B cells; pro, pro-B cells; L, large pre-B cells; S, small pre-B cells. (E) Frequency of stop codons in in-frame μMT VDJ by reading-frame. (F) Frequency of individual VH in non-productive and productive VDJ from μMT pro-B cells. (G) Frequency of individual VH in non-productive and productive VDJ from wild-type pro-B cells. (H) Frequency of individual VH in productive VDJ from wild-type pro-B cells and pre-B cells. Note: For plots (F–H), the straight line shows the line of equivalence. For plotting clarity, up to three of the lowest frequency VH were excluded from the plots, but not from the regression ($R^2$) analysis. Data Information: Data in panels B-H derived from independent VDJseq analysis of two (μMT) or three (pro-B, large pre-B, small pre-B) biological replicate pools of minimum four mice per pool, which were then merged prior to downstream analysis with RStudio. See Appendix Table S1. $R^2$ values calculated using simple linear regression in GraphPad Prism. Source data are available online for this figure.

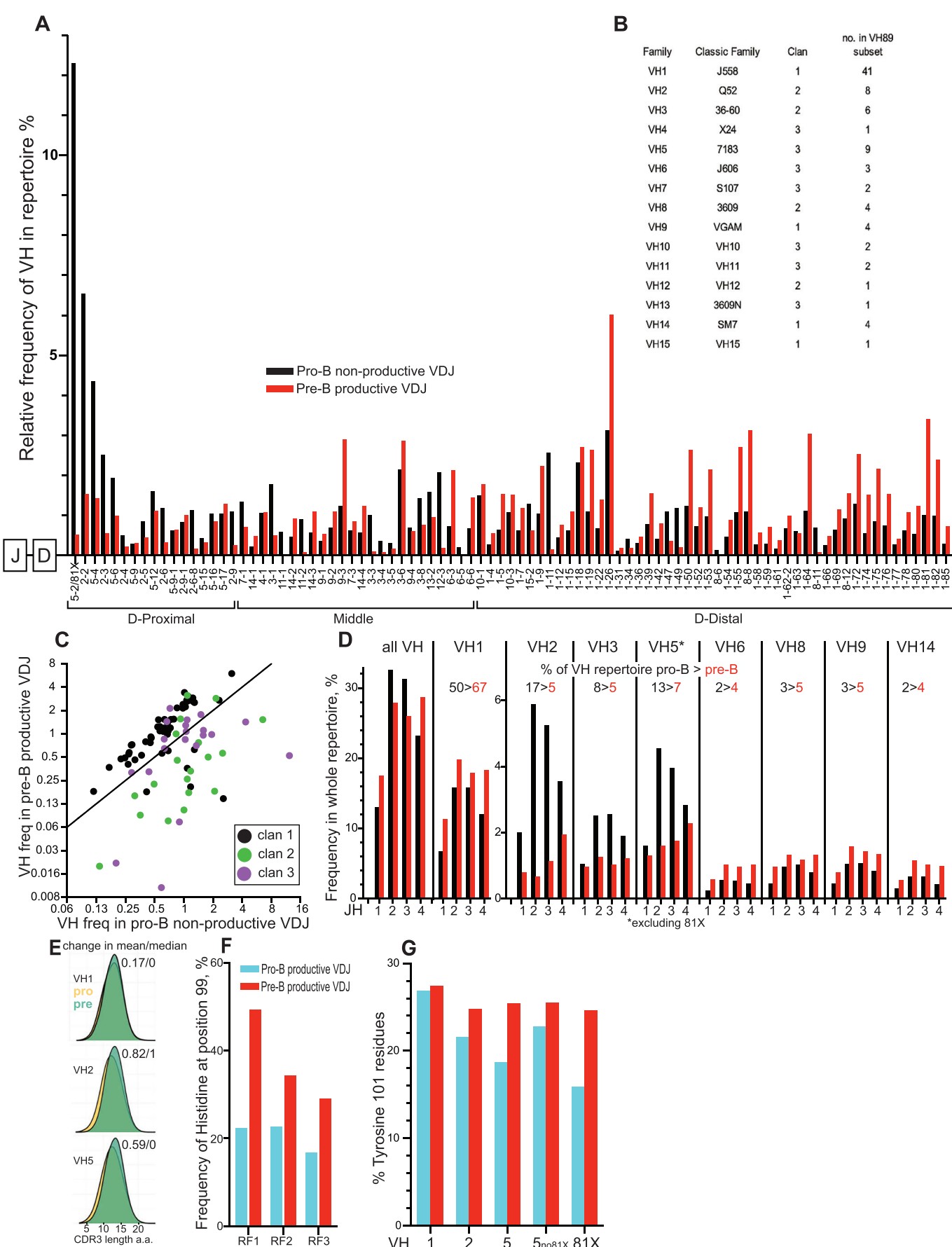

◄ **Figure 2. Selection of VH genes and families, JH and particular CDR3 amino acids, in the VDJ repertoire over the pre-B cell transition.**

(A) Change in frequency of individual VH by IgH locus position, over the pre-B cell transition, as measured from non-productive VDJ from pro-B cells to productive VDJ from small pre-B cells. (B) Table showing VH gene families, their classic names, clans, and the number of genes in the family as detected in the group of 89 most used VH. (C) Plot of frequency of individual VH in pro-B cell non-productive VDJ versus small pre-B cell productive VDJ, coloured by VH clan. (D) Frequency of particular VH-family/JH combinations as a proportion of the entire VDJ repertoire from pro-B cell non-productive VDJ and small pre-B cell productive VDJ. Numbers above bar plots refer to the overall frequency of that VH family in the VDJ repertoires measured. Note plots for VH5 exclude data from VH5-2/81X. (E) CDR3 length distribution for VH1, VH2 and VH5 families over the pre-B transition: pro, pro-B cells; pre, small pre-B cells. Numbers refer to the difference in the mean/median CDR3 length between these cell populations. Data from RF1 sequences only. (F) Frequency of Histidine residues at VH5-2/81X VDJ position 99 in the pro-B cell and pre-B cell productive VDJ repertoires, by D-reading frame. (G) Frequency of Tyrosine residues at VDJ positions 101 in the pro-B cell and pre-B cell productive VDJ repertoires, by VH-family. 5no81X, data for VH5 family excluding data for VH5-2/81X; 81X, data for VH5-2/81X alone. Data Information: Data derived from RStudio analysis of merged biological replicate datasets of VDJseq analysis of respective cell types as Fig. 1. Source data are available online for this figure.

large drop from 12.3% of the total repertoire to 0.5%. Thereafter, the trend is for the more frequently rearranged proximal VH to be strongly counter-selected, and the more distal VH to be positively selected. This confirms previous reports for small numbers of VH genes (Malynn et al, 1990), although VH3-1,12-3 and 1-11 do not follow the trend.

VH genes can be sub-divided into three clans defined by structural features conserved in evolution (Kirkham et al, 1992). Figure 2C shows that most clan 1 VH are positively selected at the pre-B transition by around twofold; 12/19 clan 2 VH show a greater than twofold counterselection; and clan 3 VH are more variably impacted.

## Strong VH-specific JH selection over the pre-B transition

JH regions are good proxies for the C-terminal part of the CDR3/junction region. The sequence they can contribute varies in length and composition (Fig. 1A). Figure 2D shows the change in frequencies of the eight largest VH families, per JH, over the pre-B transition. All VH families that show increased frequency show little differential JH selection. The three counter-selected families (VH2/3/5) show strong JH2 and JH3 negative selection and less or no selection against JH1 and JH4. JH2 and JH3 are two amino acids shorter in their CDR3 contribution (Fig. 1A), their counter-selection resulting in longer CDR3s (Fig. 2E). We suggest the extra tyrosine residues in JH1 and JH4 are closer to the λ5 sensing site providing the necessary stability for VH2/3/5 pre-BCR formation (Bankovich et al, 2007).

This surprisingly strong VH-family-specific JH selection reflects most of the negative VDJ-selection occurring at the pre-B transition. It results in removal of 21% of the repertoire, mostly JH2 and JH3 bearing proximal VH2/3/5. This contrasts with the distal VH1s which, at the level of JH, show no restriction in CDR3 diversity over the pre-B transition.

## Selection of key CDR3 amino acids over the pre-B transition

To allow direct comparisons between CDR3 amino acid sequences from pro-B and small pre-B-cells, we have compared productive VDJ from each dataset. Figure 2F confirms selection for H99 in VH5-2/81X VDJ, also showing that H99 selection occurs for VDJ in all RFs. Figure 2G shows Y101 analysis for the 3 largest VH families. VH5 shows the greatest Y101 selection, although this appears mostly due to strong counter-selection of VH 5-2/81X VDJ, that have lower levels of Y101. Excluding VH5-2/81x, the VH2 family shows the strongest selection for Y101.

Figure 3A shows overall changes in amino acid composition of the N-terminal of CDR3s over the pre-B transition. A97 and R98, already very common in V-D junctions, are further selected in all the major VH family VDJs shown. They are VH encoded, and their selection is CDR3-independent as it occurs in all CDR3 reading frames (Fig. EV4A), underlining their importance in these positions for functional CDR3s. The strong selection of A97 in VH1s follows the counter-selection of VH1-11, the only VH1 with a G97. VH1-11 falls from 2.1% to 0.15% of the total VH repertoire over the pre-B transition (Fig. 2A). H99 is selected in other VH5, in addition to VH5-2/81X, but counter-selected in VH2-family VDJ (Fig. 3A). The counter-selection of S/K98 in VH2 genes suggests, as for G97 in VH1-11, that it may be these N-terminal CDR3 amino acids, as well as perhaps other VH attributes, that inhibit pre-B transition of such VH.

## Little selection for CDR3 charge, hydropathy and aliphatic index over the pre-B transition

Positively charged Arginine (R) residues in the CDR3 were thought to be counter-selected over the pre-B transition (Keenan et al, 2008). We find no evidence for this (Fig. EV4B,F). This suggests the increased frequency of Arginine residues in CDR3s of SLC −/− mouse pre-B cells in this report, is due to reduced positive selection of CDR3s not containing Arginine. Further, we find little change in the overall CDR3 biophysical properties (Fig. EV4C–E,G–I).

## Variable Dµ selection is followed by efficient RF2 counter-selection at the pre-B transition

Figure 3A also shows that from VDJ amino acid position 100 onwards, in the CDR3, G and Y appear to be positively selected and T and V are counter-selected, particularly for VH2 and VH5. This is consistent with RF selection, as G/Y are common in RF1 and T/V are common in RF2 (Fig. 1A). RF2 counter-selection occurs prior to the pre-B transition (Fig. 1C), (Kitamura et al, 1991; Zemlin et al, 2008), although in mice expressing excess levels of RF2 VDJ, the pre-B transition is negatively impacted (Khass et al, 2016), suggesting RF2 counter-selection may also occur at this later point.

The selection of RF2, per VH, over the pre-B transition is shown in Fig. 3B. While the average change is -4% there are some striking differences in the frequency of RF2 VDJ.

In total, 39% of VH5-2 VDJs are RF2, similar to the overall 43% RF2 in µMT pro-B cells (Fig. 1C), implying little Dµ selection has occurred prior to VH5-2 recombination. Other proximal VH (VH2 and VH5) also show high RF2 proportions as do two distal VH1.

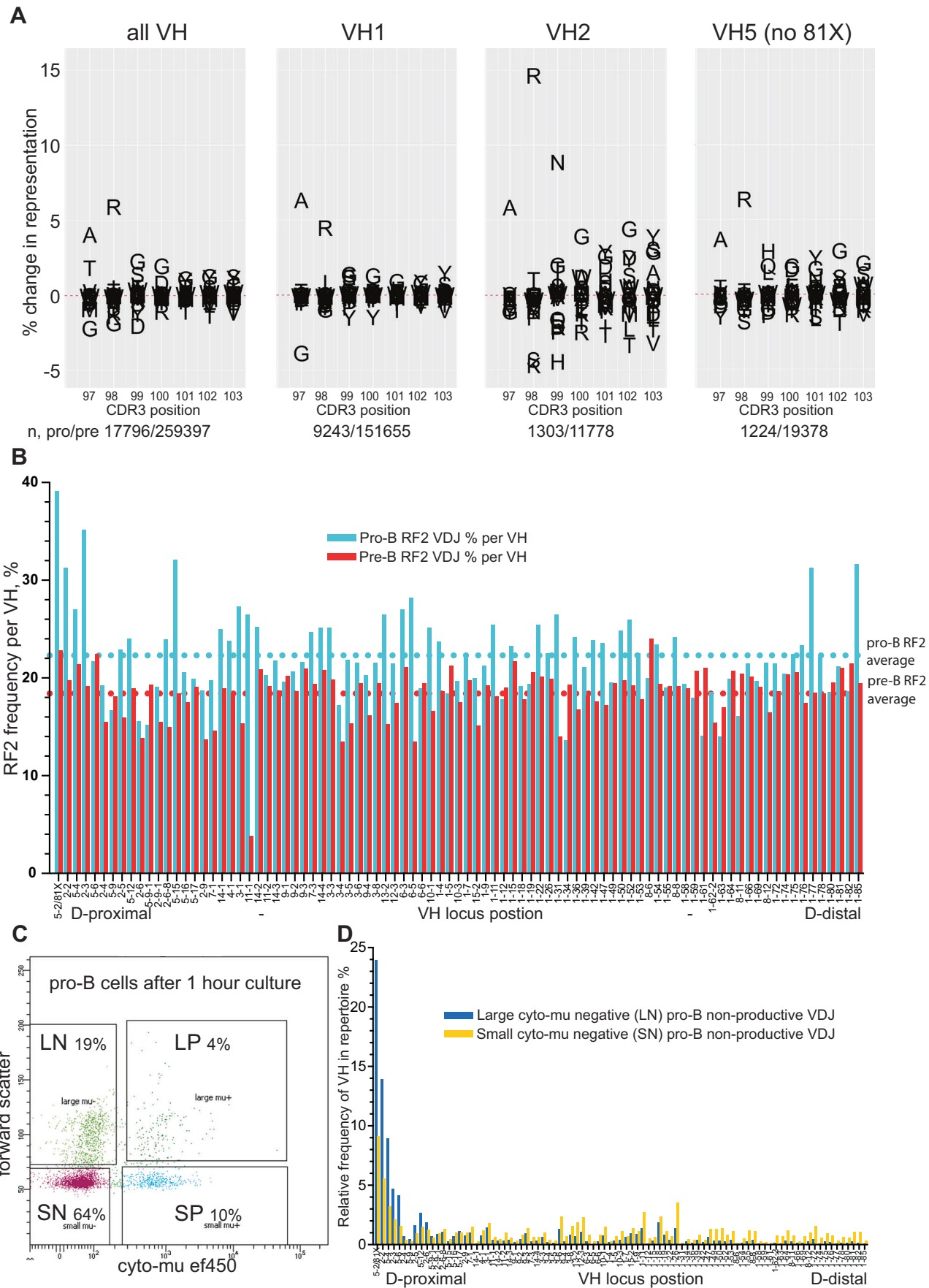

Figure 3.  Change in amino acid frequency by CDR3 position, and RF2 counter-selection, over the pre-B cell transition.

(A) Change in frequency of amino acid residue use for the three major VH families, in the first seven positions of the CDR3, over the pre-B transition as measured between pro-B cell and small pre-B cell productive VDJ. Numbers below panels show the number of sequences analysed for each VH family from pro- and pre-B cell VDJ repertoires. (B) Frequency of RF2 D-reading frame per VH over the pre-B transition, plotted by locus position, as measured between pro-B cell and small pre-B cell productive VDJ. (C) FACS plot showing separation and gating for sub-setting pro-B cells, after 1 h of culture, into four populations based on expression of cytoplasmic μ-chain and forward scatter. LN large μ-chain negative cells, SN small μ-chain negative cells, SP small μ-chain positive cells, LP large μ-chain positive cells. (D) Frequency of individual VH in non-productive VDJ repertoires from large μ-chain negative (LN) pro-B cells and small μ-chain negative (SN) pro-B cells, by locus position. Data Information: Data derived from RStudio analysis of merged biological replicate datasets of VDJseq analysis of respective cell types, as Fig. 1. Source data are available online for this figure.

All of this 'excess RF2' is eliminated at the pre-B transition, demonstrating that whether or not Dμ-mediated RF2 counter-selection has occurred, further selection at the pre-B transition is sufficient to reduce RF2 use. This explains how RF selection can occur in humans, where Dμ-selection is absent (Benichou et al, 2013; Minegishi and Conley, 2001).

We hypothesised that proximal VH recombine before Dμ-selection has had time to manifest. After 1 h of culture, pro-B cells can be subsetted into four populations based on intracellular IgM and size (Fig. 3C). One of these populations, large cyto-μ-negative cells (LN) are the least mature pro-B cells as they show the lowest frequency of V-D recombination (Appendix Table S1). Figure 3D shows the VH frequency in VDJs of these cells compared to the small cyto-μ-negative cells (SN) that represent the bulk of pro-Bs. These LN pro-B cells show a strong bias of V–D recombination to the most proximal VH, with VH 5-2 making up 24% of the total VDJ, and the most proximal 4 VH accounting for 52% of VDJs. Dμ levels need to be high to stop V-DJ recombination (Reth et al, 1985), and clearly the most proximal VH are recombining before Dμ inhibition has been established, explaining their high RF2 levels (Fig. 3B).

## Clan-specific VDJ selection prior to pre-BCR-mediated proliferation

We have shown differential selection of VH prior to pre-BCR mediated proliferation (Figs. 1G and EV2).

Figure 4A shows the level of productive VDJ for each VH by cell type. Whilst mean levels are similar between μMT and normal pro-B cells, most VH in the latter have either a higher or lower frequency of productive VDJ than the mean. This selection is strongly associated with clan and is maintained over the pre-B transition (Fig. 4B).

The association of VDJ productivity with clan is also a trend of VDJ productivity with VH locus position (Fig. 4C). The lower productivity of proximal VH VDJs arguably could be because they recombined earlier (Fig. 3D) and consequently the productive VDJ have exited the pro-B compartment before detection. However, this is not the case, as it would result in equivalent productivity levels between clans in the pre-B compartments, which we do not observe (Fig. 4B).

Importantly, most of the variation in pro-B cell VDJ productivity is absent in μMT pro-B cells (Fig. 4A), implying that this VDJ productivity variation is largely driven by heavy-chain signalling, and not intrinsic variation between VH in forming productive VDJs.

## Productive VH5-2/81X rearrangements disappear from the pre-B cell repertoire

Currently, it is thought that if a first VDJ rearrangement is productive but cannot pair with the SLC, the second allele rearranges, and if this is functional, then the cell will enter the pre-B transition with two productive heavy chains. Non-pairing μ-chains are not toxic, and such cells are thought to make up ~5% of post pre-B transition cells (Minegishi and Conley, 2001; ten Boekel et al, 1998). The data for VH5-2/81X does not agree with this model. 8.8% of productive pro-B VDJ are VH5-2 (Fig. EV3). As most of these do not pair, and the second allele is rearranged if available, these productive VH5-2 VDJ should enter the pre-B compartment as passengers. We only see 0.53% of the repertoire as productive VH5-2 VDJ in small pre-B cells (Fig. EV3), and this figure includes functional VH5-2/81X VDJ. It appears that many productive VH5-2 VDJ are 'disappearing' from the repertoire.

As most productive VH5-2/81X VDJ from pro-B cells do not pair with the SLC, and most disappear during the pre-B transition, we hypothesised that many such non-pairing VH5-2 VDJ rearrangements, and likely others, become targets for VH replacement and that this explains their disappearance as compared to non-productive VDJ.

## Low VDJ productivity correlates with possession of a cryptic RSS

Cryptic-RSSs facilitate VH-replacement, so we analysed their association with VDJ productivity. Low VDJ productivity correlates strongly with the presence of a cRSS in the VH (Fig. 4D). This picture is complicated, however, since many VH with a cRSS have higher productivity VDJ. This suggests possession of a cRSS is necessary but not sufficient to render a VDJ vulnerable to VH-replacement, and other factors are important, for example, the frequency of formation of non-pairing VDJ. There are a few VH without cRSS that generate low productivity VDJ (Fig. 4D). It is notable, however, that the two VH without cRSS with the lowest productivity rates (VH11-1/11-2), marked with blue dots, have V-segments that end with the first two basepairs of a stop codon (TA). This suggests these VH, along with the single other VH ending TA (VH3-8, indicated in Fig. 4D), have an intrinsic tendency to form non-productive VDJ. Overall, these results are consistent with VH replacement of certain productive VDJ in pro-B cells, mediated by recombination at the cRSS. In contrast to productive VDJ, no non-productive VDJ shows strong VH-selection over the pre-B transition, Fig. 4E.

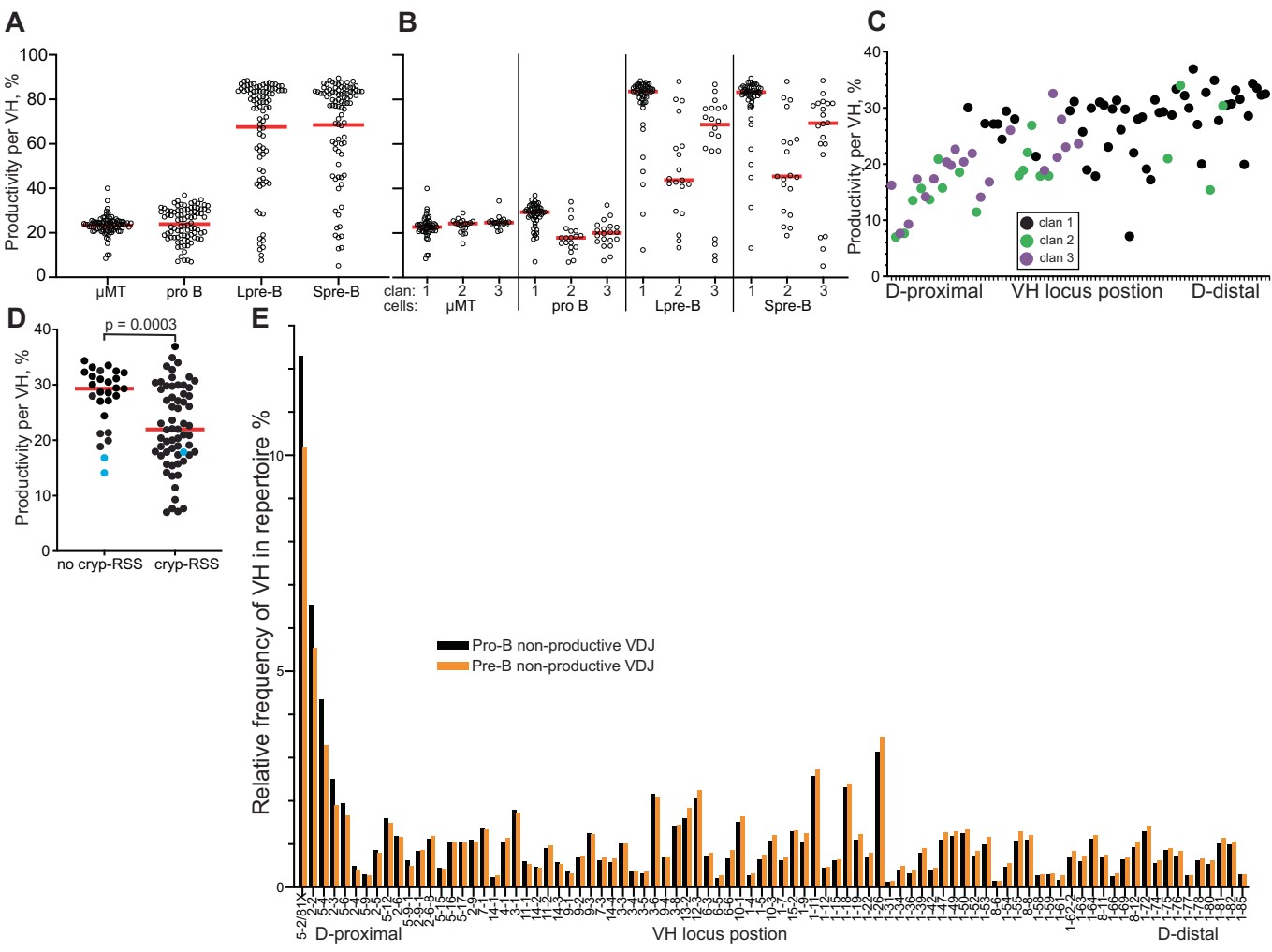

**Figure 4. Selection and counter-selection of productive VDJ over the pre-B transition.**

(A) VDJ productivity per VH for the four cell types analysed. μMT, μMT pro-B cells; pro B, pro-B cells; Lpre-B, large pre-B cells; Spre-B, small pre-B cells. Red bars show mean values. (B) VDJ productivity per VH by VH clan, for the four cell types analysed. x-axis labels as previous panel. Red bars show median values. (C) Pro-B cell VDJ productivity per VH plotted by IgH locus position and coloured by clan—see key. (D) VDJ productivity per VH for pro-B cell VDJ whose VH do and do not have a cryptic-RSS. Blue dots indicate the three VH that have sequences ending in TA as the first two basepairs of a codon, increasing their probability of generating a stop codon. no cryp-RSS, no cryptic RSS; cryp-RSS, has cryptic RSS. Red bars show median values. Cryptic RSS defined as TACTGTG, a few basepairs from 3′ of V-segment. (E) Frequency of VH in non-productive VDJ from pro-B and small pre-B cells by locus position. Data Information: Data derived from RStudio analysis of merged biological replicate datasets of VDJseq analysis of respective cell types, as Fig. 1. Significance testing was performed with an unpaired two-tailed *t* test with Welch's correction in GraphPad Prism. Source data are available online for this figure.

## VH-replacement is biased toward productive non-pairing VDJ rearrangements

In VHR, the intervening sequence in the *Igh* locus loops out and can form an 'excision circle' after resolution of the event (Usuda et al, 1992). In contrast to normal VDJ-rearrangement excision-circles, where back-to-back RSSs are flanked by intergenic DNA, in VHR, cleavage at the cryptic RSS could produce an excision circle with the replaced VH adjacent to its cRSS, ligated to the RSS and 3′ intergenic sequence of the invading VH (Fig. 5A). We designed a sequencing strategy, 'RCseq' to detect VHR circles, with the run-off primer position indicated on Fig. 5A, 'Seq.'

To avoid sequencing every VH from every cell, the sequencing primers focused on VH more likely to have been replaced, i.e.,

those that show a large drop in productive VDJ levels. These were VH5-2/4/6/12 and VH2-2/3/6/-6-8; and VH3-1 and VH1-11. VH1-11 shows the largest drop in productive VDJ for VH1s, which generally show strong positive selection (Fig. 2A). As controls, we included primers targeting the positively selected VH3-6, for comparing to VH3-1; and VH1-26 and 1-81 for comparing to VH1-11.

We found 53 sequences in total in gDNA from 3 pro-B cell pools and a large pre-B cell pool. All sequences that fulfilled the above search criteria (VH followed by back-to-back RSS (+/− insertions) followed by ectopic VH intergenic sequence), showed donor genes distal to recipients, as would be expected if the circles were generated by VHR. The VH replacement circle sequences are in Appendix Table S3, and donor/recipient pairs are in Appendix

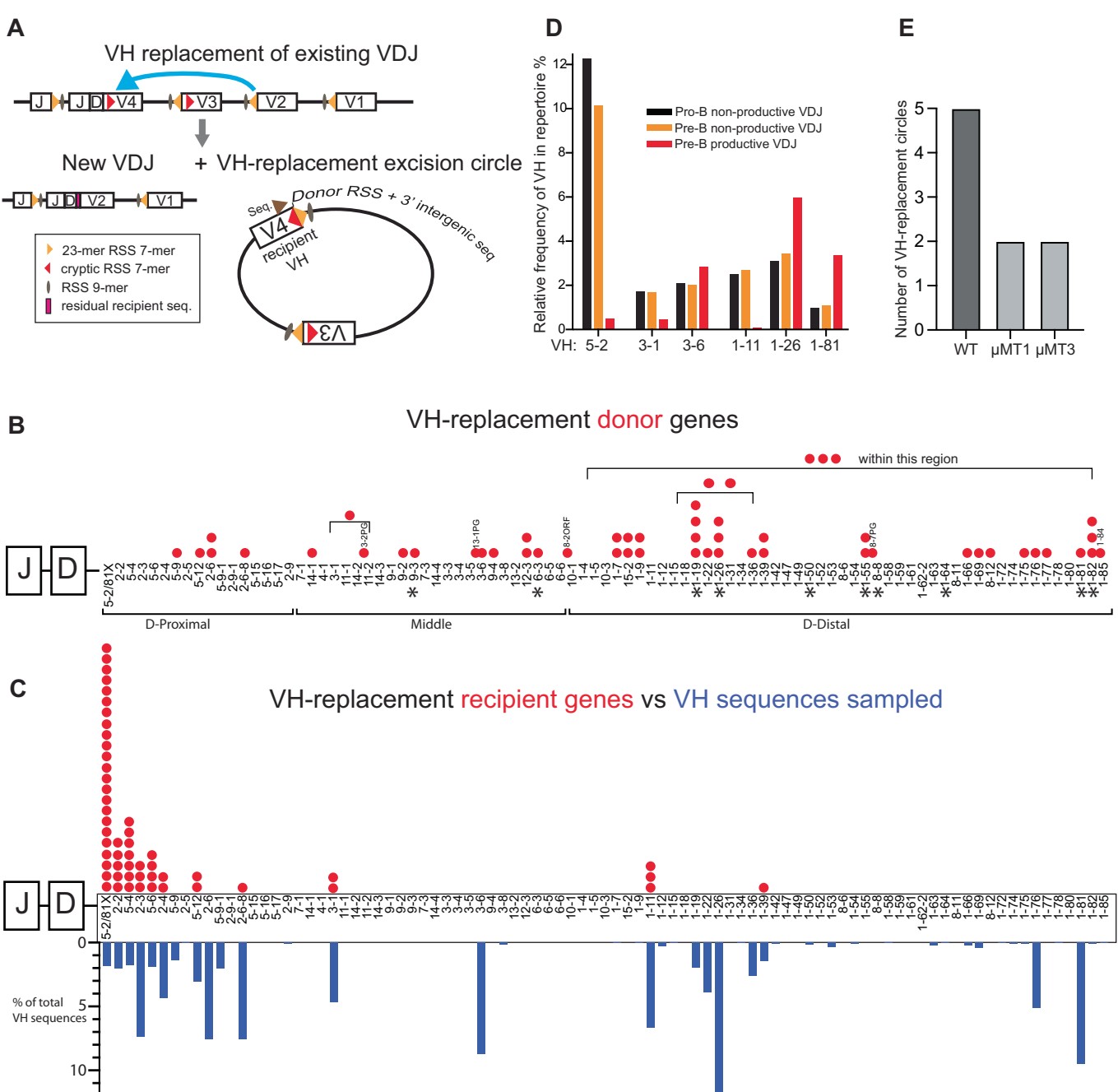

© The Author(s)

Table S4. In all, 18/53 sequences have insertions of between 1 and 4 bp between the back-to-back 7mer RSSs.

The position of the donor/invading VH genes on the VH locus is shown in Fig. 5B. The sequencing strategy is agnostic of the identity of donor genes (Fig. 5A). VH replacement donor VH are derived from throughout the *Igh* locus, but with a bias toward distal VH1. The ten VH showing the highest gain in productive VDJ (from Fig. 2A) over the pre-B transition are marked with asterisks. We detect VHR donors from seven of these, associating VH replacement donors with high net gains in productive VDJ. Donor VH also include at least three pseudogenes and an open reading frame (ORF), VH8-2.

The position of the VH replacement recipient genes is shown in Fig. 5C. The detection of these is non-exhaustive as we only probed for the candidates described above, with primers that sampled sequences in the proportions shown in blue bars. VH5-2/81X, which shows the highest loss of productive VDJ, almost all of which cannot pair with the SLC (ten Boekel et al 1997), dominates as a target for VH replacement. The next four proximal VH, which also show high losses of productive VDJ, are also frequently replaced, compared to the other proximal VH sequenced. Importantly, for VH3-1 and VH1-11, which show large losses of productive VDJ, but no losses of non-productive VDJ (Fig. 5D), we detected 2 and 3 replacement events, but none for the controls VH3-6 and VH1-26/81, which gain productive VDJ at the

◀  **Figure 5.   Location and frequency of use of VH-replacement donor and recipient VH.**

(A) Diagram showing how VH replacement can generate an excision circle containing the replaced VH followed by the RSS and 3′ intergenic sequence of the invading/donor VH. 'Seq.' with brown arrow indicates approximate location of run-off primers for sequencing of excision circle to yield sequence that detects recipient and donor VH. (B) Diagram of IgH locus indicating position and frequency of use of VH replacement donor VH genes, from the 53 VH replacement excision circles detected. Red dots indicate detection of a donor sequence from that VH. The brackets indicate the range of locations some of the donor VH may have, due to insufficient sequence data to precisely locate them. Asterisks indicate the ten VH that show the largest gains in productive VDJ over the pre-B transition. Some of the donor VH were open reading frames (ORFs) or pseudogenes (PG), these are labelled above the dots indicating their locus position. In addition, there is one VH donor, VH1-84, that is also indicated at the extreme distal end. It is not included in the main VH locus labelling because it is not one of the 89 most frequently used VH. (C) Diagram of IgH locus indicating the position and frequency of use of VH replacement recipient VH genes, from the 53 VH replacement excision circles detected. Red dots indicate VH replacement events at that VH. The blue bars below the locus labelling indicate the VH sampling distribution of the total sequences performed. (D) Frequency of selected VH in pro-B non-productive, pre-B non-productive and pre-B productive VDJ. Data extracted from Figs. 2A and 4E. (E) Number of VH-replacement circles detected in identical amounts (540 ng) of DNA from pools of wild-type or μMT pro-B cells. WT, wild-type; μMT1, μMT pro-B cell pool 1; μMT3, μMT pro-B cell pool 3. Data Information: (B, C) Data derived from sequence analysis of RCseq (replacement-circle sequencing) datasets from three pro-B cell pools and one large pre-B cell pool; $n = 4$, data then merged. Panel D data derived from RStudio analysis of merged biological replicate datasets of VDJseq analysis of respective cell types, as Fig. 1. $n$ as for Fig. 1. (E) Data derived from sequence analysis of RCseq datasets from one wild-type pro-B cell pool and two μMT biological replicate pro-B cell pools. Source data are available online for this figure.

pre-B transition. We conclude that VHR is biased toward the removal of productive non-pairing VDJ from the pro-B cell repertoire.

If VH-replacement is driving the removal of productive non-pairing VDJ in wild-type pro-B cells, explaining the early VDJ selection we observe, and this selection is largely absent in μMT pro-B cells (Fig. 4A), it follows that such VH-replacement is enhanced by μ-chain signalling. If this is the case, we should find fewer VH-replacement circles in μMT pro-B cells. Due to differences in μMT pro-B cell VDJ, described in 'Methods' section, we could only search for replacement-circles using the shell-script search for back-to-back RSS. In our hands, this detects around half of the circles we would find by additionally using the more sophisticated search, described in 'Methods'. We analysed RCseq libraries made from identical amounts of DNA (540 ng) from one wild-type pro-B cell pool and two μMT pro-B cell pools. We found five VH-replacement circles in the wild-type pro-B-cell sample and only two in each of the μMT pro-B cell samples (Fig. 5E). The replacement-circle sequences are shown in Appendix Table S3. This result supports the proposal that μ-chain signalling enhances VH-replacement.

## Discussion

Tying VH-Replacement to removal of non-pairing VDJ rearrangements defines a new physiological role for VHR, and addresses a long-lasting question fundamental to VDJ repertoire formation, that of what happens to the non-pairing VDJ (Marshall et al, 1996; ten Boekel et al, 1997). This also explains at least some of the distal repertoire shift, as we show that distal VH are more tolerant of CDR3 composition, and thus could replace the VH in productive non-pairing proximal VDJs, making functional VDJs that increase the representation of distal VH in the repertoire.

VHR was originally found in B-cell lines (Kleinfield et al, 1986; Reth et al, 1986), and can occur more than once on the same allele, sometimes introducing charged residues to the CDR3 (Zhang et al, 2003). Through the use of genetically modified mice it has been shown to occur to productive and non-productive VDJ rearrangements (Sun et al, 2015; Taki et al, 1995). Whilst it was thought to mediate receptor editing in immature B-cells (Chen et al, 1995), this has subsequently been disproven (Sun et al, 2015), following on from observations that VHR-associated cRSS cleavage products were confined to pro-B cells (Davila et al, 2007).

While our strategy cannot determine whether the replaced VDJ was productive or non-productive, VH5-2/81X, VH3-1 and VH1-11, for which we detect VHR, principally demonstrate loss of productive rather than non-productive VDJ over the pre-B transition (Fig. 5D). This contrasts with the other VH analysed, that show no loss of productive VDJ and no evidence of VHR.

The large drop in productive VH5-2/81X VDJ over the pre-B transition, the great majority of which do not pair with the SLC, and which account for half of the VHRs we detect, is consistent with VH-replacement with a strong bias toward productive non-pairing VDJ. VH1-11, which shows a large drop in productive VDJ over the pre-B transition, uniquely encodes a G97. Supposing G97 interferes with SLC-pairing, we counted the frequency of it in non-VH1-11 pro- and pre-B cell productive VDJs. These will have appeared presumably through imprecise V-D joins. We found the frequency of G97 in non-VH1-11 VDJ drops by around two-thirds in pre-B-cells (from 144/19115 to 719/259018, 0.753% to 0.277%). This suggests that the G97 interferes with SLC-pairing in VH1-11 VDJ making it more likely to be replaced as compared to other VH1 VDJ (e.g. Fig. 5C,D).

Whilst VDJ that encode non-pairing μ-chains may be no better molecular substrates for VHR than non-productive VDJ, they may be more available for secondary recombination. Non-paired μ-chains can still signal allelic exclusion (Shimizu et al, 2002), whilst at the same time incompletely suppressing Rag and TdT (Galler et al, 2004). This would favour VH replacement over second allele recombination. In contrast, non-productive VDJ may more rapidly stop continued recombination on the same allele through nonsense-mediated decay signalling (Fuxa et al, 2004; Roldan et al, 2005), and progress to recombination on the other allele, if available, or cell death. In support of this proposed mechanism is that μMT mice undergo less of the VDJ selection (Fig. 4A) that we show is impacted by VHR, implying that this selection is enhanced by μ-chain signalling. If this is the case, we should find fewer VH-replacement circles in μMT mouse pro-B-cells, and Fig. 5E shows data supporting this hypothesis. We found a 2.5-fold higher frequency of VHR circles from a pool of WT mouse pro-B cells as compared to two pools of μMT mouse pro-B cells. Of additional note is that 4/5 of the replacement circles in WT mice showed VH5-2/81X replacement, whereas 0/4 of the replacement circles from μMT mice contained VH5-2/81X (Appendix Table S3).

μ-chains vary in their dependency on SLC components for effective signalling (Kohler et al, 2008). Non-paired μ-chains may

well also vary in their residual signalling capacity. Some can signal autonomously (Galler et al, 2004) and may support developmental progress, such as the auto-reactive heavy chains found in SLC −/− mice (Keenan et al, 2008). Some heavy-chain only antibodies can even appear in the periphery (Zou et al, 2007). However, most unpaired μ-chains will signal less than a functional pre-BCR, at least temporarily stalling the cell, giving more opportunity for rescue by VHR.

On this note, the study of Davila et al (Davila et al, 2007) reported comparable efficiencies of recombination, using an in vitro assay, between various other cRSSs located upstream and the 3′ cRSS. VHR involving these would generate larger, likely non-functional VDJ polypeptides. This raises the possibility that VHR occurring away from the 3′ cRSS can also release pro-B cells stalled by a non-pairing μ-chain.

Our detection of VHR greatly underestimates the number of events. Firstly, most VHR will result in non-productive VDJ. This will lead to cell death or second allele rearrangement with similar chance of failure. Secondly, our detection of VHR is dependent on detection of ligated excision circles. Even the fate of the un-ligated signal joint ends in normal V-DJ recombination is unclear (Lieber, 2010). Thirdly, if VHR is occurring at sites other than the most 3′ cRSS, our assay would not detect it. In addition, up to 20% of peripheral B-cell VDJ can be derived from VHR in a mouse with a monoclonal, functional VDJ, despite a shortened pro-B cell transit (Kumar et al, 2015; Sun et al, 2015). That such high levels of VHR can occur, suggests it can account for the changes in productive VDJ levels we observe in pro-B cells prior to pre-BCR-mediated proliferation (Fig. 1G), especially considering the likely extended pro-B phase of cells with non-pairing μ-chains. Also, cells with VHRs that generate pairing μ-chains will rapidly exit the pro-B compartment, suggesting some of the selection manifesting after pre-B cell proliferation is generated by VHR rather than differential proliferation.

That we detect VH-Replacement donor VH from right across the *IgH* locus, in a physiological context, is also consistent with a developmental stalling of B-cells with productive non-pairing VDJ. Such stalling would allow the locus contraction necessary to facilitate distal VH to be donors for VH-Replacement of a more proximal VDJ. In contrast, a mouse knock-in model using a functional VDJ showed only proximal VH as donors for VH-Replacement, consistent with an accelerated pro-B cell transit (Sun et al, 2015).

We have shown independent selection of three CDR3 sequence features over the pre-B transition: N-terminal amino acids, RF, and strong VH-specific counter-selection of JH. The selection of the VH encoded N-terminal A97 and R98 is VH-universal, highlighting the importance of these residues for successful pre-B transition, as is the counter-selection of RF2. Conversely, the strong VH-specific counter-selection we observe, that removes a third of the pro-B cell VDJ repertoire, is associated either with JH2/3 in VH2/3/5 or with specific CDR3 residues, e.g., G97 in VH1-11 or S/K98 in VH2s. We suggest these features impact SLC-pairing the most, driving most of the VH-replacement, excepting for VH5-2/81X, which largely fails to pair.

B-cell cytokine signalling is altered in ageing and chronically inflamed bone marrow (Dowery et al, 2021; Koohy et al, 2018; Pioli et al, 2019). Inflammatory stimuli can drive the release of B-cell progenitors into the periphery (Nagaoka et al, 2000; Ueda et al, 2004). These stresses significantly impact cells undergoing the pre-B transition, re-locating them to the spleen. This is highly likely to have impacts on the peripheral heavy-chain repertoire. Our in-depth study of the structure and selection of the normal heavy-chain repertoire provides a good foundation to further investigate such alterations.

# Methods

### Reagents and tools table

| Reagent/resource | Reference or source | Identifier or catalogue number |
|---|---|---|
| **Experimental models** | | |
| C57BL/6JBabr (mus musculus) | Babraham Inst. | n/a |
| μMT (mus musculus) | Babraham Inst. | n/a |
| **Recombinant DNA** | | |
| n/a | | |
| **Antibodies** | | |
| Biotin anti-mouse CD11b | eBioscience | 13-0122-85 |
| Biotin anti-mouse Ly6G | eBioscience | 13-5931-85 |
| Biotin anti-mouse Ter-119 | eBioscience | 13-5921-85 |
| Biotin anti-mouse CD3e | eBioscience | 13-0033-86 |
| Anti-CD19 MACS beads | Miltenyi | 130-121-301 |
| Anti-mouse CD16/32 Fc block | eBioscience | 14-0161-86 |
| BV421 anti-mouse CD19 | BD Biosciences | 115538 |
| BV650 anti-mouse CD93 | BD Biosciences | 752943 |
| PE-Cy5 anti-mouse CD25 | eBioscience | 15-0251-82 |
| Ef660 anti-mouse IgM | eBioscience | 50-5790-82 |
| PE anti-mouse IL7-Ra | eBioscience | 12-1273-83 |
| PE-Cy7 anti-mouse CD43 | BD Biosciences | 562866 |
| FITC anti-mouse CD24 | eBioscience | 11-0241-85 |
| APCefluor780 viability dye | eBioscience | 65-0865-14 |
| BUV737 anti-mouse CD19 | BD Biosciences | 612781 |
| Ef450 anti-mouse IgM | Invitrogen | 48-5890-82 |
| **Oligonucleotides and other sequence-based reagents** | | |
| VDJseq primers | Ref: Chovanec et al, 2018 | Tables 1, 2, 3, 4 |
| RCseq primers | This study | Appendix Table S5 |
| **Chemicals, enzymes and other reagents** | | |
| Quick DNA mini-prep plus kit | Zymo | D4068 |

| Reagent/resource | Reference or source | Identifier or catalogue number |
|---|---|---|
| QIAquick PCR purification kit | Qiagen | 28106 |
| Agarose | Cleaver Scientific | CSL-AG500 |
| 100 mM dNTPs | NEB | 55082 to 55085 |
| T4 DNA polymerase | NEB | M0203 |
| T4 polynucleotide kinase | NEB | M0201 |
| DNA large pol 1 Klenow | NEB | M0210 |
| Ultra II end-prep module | NEB | E7646A |
| Ultra II ligation module | NEB | E7648A |
| Klenow fragment 3'-5' exo- | NEB | M0212 |
| T4 DNA ligase | NEB | M0202 |
| Vent exo- DNA polymerase | NEB | M0257 |
| Dynabeads MyOne streptavidin T1 beads | Invitrogen | 65601 |
| Tris-HCl pH 7.5 | Sigma | T2319-1L |
| EDTA 0.5 M | Invitrogen | 15575-038 |
| NaCl 5 M | Sigma | 71386-1 L |
| Tween-20 | Fisher BioReagents | BP337-100 |
| Sodium Acetate 3 M | Sigma | S7899-500ML |
| Q5 High-fidelity PCR master mix | | M0492 |
| Absolute ethanol | VWR | 20821.330 |
| Agencourt AMPPure XP beads | Beckman Coulter | A63880 |
| Oligonucleotides | Sigma Merck | n/a |
| NEBnext Library Quant Kit | NEB | E7630 |
| Foetal bovine serum | Bio Sera | FB-1001/500 |
| PBS | Sigma | D8537-500ML |
| RPMI medium | Sigma | R8758-500ML |
| HEPES buffer | Gibco | 15630-056 |
| DNA/RNA Shield | Zymo | R1100-50 |
| Streptavidin Microbeads | Miltenyi | 130-048-101 |
| **Software** | | |
| Fast QC | http://www.bioinformatics.babraham.ac.uk/projects/fastqc/ | |
| Trim Galore! | http://www.bioinformatics.babraham.ac.uk/projects/trim_galore/ | |
| Cutadapt | https://cutadapt.readthedocs.org/en/stable/ | |

| Reagent/resource | Reference or source | Identifier or catalogue number |
|---|---|---|
| Bowtie 2 | http://bowtie-bio.sourceforge.net/bowtie2/index.shtml | |
| PEAR | http://www.exelixis-lab.org/web/software/pear | |
| Kalign2 | http://msa.sbc.su.se/ | |
| SAMtools | http://samtools.sourceforge.net/ | |
| IgBlast | https://www.ncbi.nlm.nih.gov/igblast/faq.html#standalone | |
| BabrahamLinkON | https://github.com/peterch405/BabrahamLinkON | |
| RCseq tools | https://github.com/s-andrews/replacementcircles | |
| **Other** | | |
| Illumina MiSeq | Illumina | |
| Aviti sequencer | Element Biosciences | |

## Mice

Wild-type C57BL/6Babr and μMT mice were maintained in the Babraham Institute Biological Services Unit in accordance with the institute's Animal Welfare and Ethical Review Body and Home Office Rules under Project License 80/2529. ARRIVE guidelines were followed. All mice were males between 11.7 and 14 weeks old.

## Primary cells

Mice were euthanised with $CO_2$ asphyxiation followed by cervical dislocation, or cervical dislocation followed by pithing. Wild-type bone marrow was flushed from femurs and tibias, using 25 G needles with RPMI medium/5% FBS/24 mM HEPES, washed once with PBS and subject to non-B cell depletion in PBS/2 mM EDTA/0.5% FBS, using biotinylated antibodies against CD11b (MAC-1; ebioscience; 1:1600), Ly6G (Gr-1; eBioscience; 1:1600), Ter119 (ebioscience; 1:400) and CD3e (ebioscience; 1:800), followed by incubation on ice for 30 min. Cells were washed and Streptavidin MACS beads were then added (5 μl/$10^7$ cells in 100 μl; Miltenyi) and incubated at 4 deg for 15 min with occasional mixing. MACS LS columns were equilibrated and washed cells were loaded with the flow-through collected for flow sorting. After flushing μMT mouse bone marrow cells were washed once with PBS and subject to positive selection for CD19 using MACS LS columns (Miltenyi) according to the manufacturer's instructions. Cells were then used directly for genomic DNA extraction.

## Flow cytometry

Mouse bone marrow (BM) B-cell progenitors were sorted from femurs and tibias, for at least two biological replicate pools of 4 or more, 12–14-week-old, C57BL/6J male mice (Fig. EV1A). B-cell

enriched wild-type bone marrow cells were resuspended in PBS/2.5% FBS/2 mM EDTA and Fc Blocked for 10 min (1:400, eBioscience, clone 93) then stained for flow sorting with the following markers: BV421 anti-CD19 (1:400, BD Biosciences, clone 1D3), BV650 anti-CD93 (1:800, BD Biosciences, cloneAA4.1), PE-Cy5 anti-CD25 (1:500, eBioscience, clonePC61.5), ef660 anti-IgM (1:400, eBioscience, clone11/41), PE anti-IL7Ra (1:400, eBioscience, clone eBioSB/199), PE-Cy7 anti-CD43 (1:100, BD Biosciences, cloneS7), FITC anti-CD24 (1:800, eBioscience, clone 30F1), APCeFluor780 viability dye (1:2000, eBioscience). When staining for cytoplasmic μ-chain was also done, cells were stained according to the above panel but using BUV737 anti-CD19 (1:400, BD Biosciences, clone 1D3). Cells were washed, fixed using Cytofix/Cytoperm (BD Biosciences) and stained for 30 min on ice with ef450 anti-IgM (1:400, Invitrogen, clone eb121-15F9). Cells were sorted on a BD FACSAria Fusion or FACSAria II. The flow sorter gating strategy is shown in Fig. EV1A. The gating strategy combines and extends previous strategies. The phenotypes were: All cells, CD19+, CD93+; pro-B cells (Hardy fraction BC, Basel pre-BI), CD25−, IgM−, IL-7R+, CD43hi, CD24mid; large pre-B cells (Hardy fraction C', Basel large pre-BII) CD25−, IgM−, IL-7R+ CD43mid/lo, CD24hi; small pre-B cells (Hardy fraction D, Basel small pre-BII), CD25+, IgM−. We have analysed c-kit+ (clone 2B8) CD43+, IgM- pro-B cells and find some c-kit+ cells have a large pre-B cell phenotype (Fig. EV1B). Pro-B subset gates, using cytoplasmic μ-chain staining, are shown in Fig. 3C. Cells were then centrifuged at $450 \times g$ for 5 min, resuspended in 1× DNA/RNA shield and frozen at −20 °C.

## VDJseq

Mouse bone marrow (BM) B-cell progenitors were sorted from femurs and tibias, for at least two biological replicate pools of 4 or more, 12–14-week-old, C57BL/6J male or μMT male mice (see Appendix Table S1).

Genomic DNA from individual progenitor cell pools was subject to quantitative VDJ repertoire analysis as previously described in a step-wise protocol (Chovanec et al, 2018). This detects both DJ and VDJ recombinants allowing measurement of the frequency of VDJs in populations.

Libraries were sequenced on an Illumina MiSeq. Sequence reads were analysed using the Babraham Linkon pipeline (https://github.com/peterch405/BabrahamLinkON).

After processing sequence data using the BabrahamLinkon pipeline, fasta files from the penultimate step were also processed through the IMGT/HighV-QUEST web service to extract D-region reading frames, which were then merged with the main data using sequence ID in R. We use the D reading-frame convention of Ichihara et al (Ichihara et al, 1989), not IMGT, so RF i.d's were transposed 1 to 2, 2 to 3, 3 to 1.

Of the 195 C57BL/6 VH genes, 123 are recombinationally active (Bolland et al, 2016). We focused on VH recombinants detected at >0.1% frequency, to allow reproducible comparisons from datasets of varying sizes. These 89 VH (Appendix Table S2) represent ~98% of productive VDJ recombinants in a pro B-cell repertoire.

To address a few minor discrepancies in the IMGT VH gene reference database, we have used a slightly modified VH reference database in the BabrahamLinkon pipeline used to analyse VDJseq data, which can be called using custom_ref during clone assembly.

https://github.com/peterch405/BabrahamLinkON/tree/master/babrahamlinkon/resources/IgBlast_database.

We observed strong correlation of VH frequency between biological replicates ($R^2 = 0.98$–0.99), (Fig. EV1C), so replicate sequence pools were merged.

### Statistics

All downstream analysis was performed using RStudio to generate spreadsheets and/or data suitable for use in GraphPad Prism, which was used for plotting and statistical analysis. Correlation was measured with simple linear regression. This is considered a standard approach for testing the strength of a linear relationship between two continuous variables. Significance testing was performed with an unpaired two-tailed $t$ test with Welch's correction.

We consider that VDJseq does not introduce subjective bias in the analyses of different cell phenotypes. Both the NGS and the bioinformatic analysis are standardised and machine-run. Blinding of samples was not considered necessary.

## Use of μMT pro-B cells

Despite extensive back-crossing to C57BL/6 mice, the 129-mouse-derived μMT *Igh* locus is likely slightly different from that of C57BL/6. Nevertheless, VDJseq reproducibly detects 88 of the 89 VH we analyse, in biological replicates of μMT pro-B cell pools (Fig. EV1C, $R^2 = 0.99$). Whilst there is also strong correlation in VH frequency between C57BL/6 and μMT non-productive VDJs, however, there is still some variation (Fig. EV1D, $R^2 = 0.76$). For this reason, we are not directly comparing VH frequencies between these strains, but using μMT mice to analyse other fundamentals of VDJ recombination.

## VH replacement excision circle sequencing

Excision circles were sequenced using an adapted VDJseq protocol. We used nested VH framework-3 primer pools for the single-cycle run-offs, followed by the first PCR step of the VDJseq protocol. Sequences shown in Appendix Table S5. These targeted VH2-2,2-3,2-6,2-6-8,5-2,5-4,5-6,5-12,1-11,1-26,1-81,3-1,3-6. The run-off annealing temperature was changed to 56 °C, and the PCR1 annealing temperature to 60 °C, with 12 or 13 cycles depending on starting DNA amounts. Library preparation and sequencing then resumed as for VDJseq. Libraries were sequenced on an Illumina MiSeq or an Element Biosciences Aviti.

Most sequences originate from the germline VH genes present in every cell. If VH replacement occurs and resolves in a manner related to normal VDJ recombination, there should be VH-replacement circles, with VH sequence followed by a head-to-head 7-mer followed by ectopic intergenic sequence 3′ from the invading VH.

Raw sequences were searched for VH replacement circles in two ways. The main approach was to assemble the paired-end reads using PEAR, reverse and complement them, and deduplicate using the UMIs and sequence length. We then ran sequences through a bespoke script to detect replacement circles. This searches for 60 bp of framework-3 sequence of any particular VH followed by a variable gap that includes the back-to-back 7-mer RSSs, which allows for imprecise joining, and then for 60 bp of the intergenic

sequence 3′ of any VH that is not the same as the framework-3 VH. These scripts are available here: https://github.com/s-andrews/replacementcircles Sequences were then manually curated to confirm the presence of the back-to-back cryptic 7-mer/normal 7-mer RSSs and confirm the identity of the VH sequences involved using IgBlast. As an adjunct, to search reads that didn't extend to the end of the second VH search region, we also searched deduplicated reads directly for back-to-back cryptic 7-mer/normal 7-mer RSS sequences, using grep in shell script, allowing for up to 3 bp insertions between RSSs. The most common back-to-back cryptic 7-mer/normal 7-mer RSS sequence is TACTGTG/CACAGTG, although there is variation in the normal VH RSS 7-mer necessitating a redundancy in this search, TACTGTG/CACA[-GAT]TG. Returned sequences not present from the main search were manually curated in the same way.

## Data availability

The VDJ-seq and VH-replacement circle raw sequencing files generated in this study have been deposited in SRA under Bioproject ID PRJNA1194793 and are publicly available. The minimally processed VDJseq datasets in AIRR format are publicly available, for the four cell types (pro-B, large pre-B, small pre-B, μMT) from Zenodo https://doi.org/10.5281/zenodo.15584842.

The source data of this paper are collected in the following database record: biostudies:S-SCDT-10_1038-S44318-025-00552-8.

## Peer review information

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

## Acknowledgements

The authors thank all the members of the Babraham Institute Flow Cytometry facility, Biological Services Unit, Bioinformatics Group and Genomics Facility. We thank Alex Whale and Jon Houseley for valuable discussion. We also thank Martin Turner for reviewing the manuscript. PC was supported by a Biotechnology and Biological Sciences Research Council, UK (BBSRC) Industrial CASE PhD studentship award (1520397). GB was supported by a joint Babraham Institute/University of Cambridge PhD studentship. EF was supported by a Medical Research Council University of Cambridge Doctoral Training Fellowship. Research in AEC's laboratory was supported by grants from the BBSRC, UK (BBS/E/B/000C0427, BBS/E/B/000C0428) and the UKRI-BBSRC Core Capability Grant funded Babraham Institute facilities.

## Author contributions

**Harry N White**: Conceptualisation; Data curation; Software; Formal analysis; Validation; Investigation; Visualisation; Methodology; Writing—original draft; Project administration; Writing—review and editing. **Peter Chovanec**: Software; Methodology; Writing—review and editing. **Laura Biggins**: Data curation; Software; Formal analysis; Visualisation. **Elise C French**: Investigation. **Georgia Bullen**: Investigation. **Simon Andrews**: Conceptualization; Data curation; Software; Formal analysis. **Anne E Corcoran**: Conceptualization; Supervision; Funding acquisition; Validation; Project administration; Writing—review and editing.

Source data underlying figure panels in this paper may have individual authorship assigned. Where available, figure panel/source data authorship is listed in the following database record: biostudies:S-SCDT-10_1038-S44318-025-00552-8.

## Disclosure and competing interests statement

The authors declare no competing interests.

# Expanded View Figures

**Figure EV1. FACS gating strategy and VDJseq library replicate comparisons.**

(A) Babraham Institute FACS gating strategy for selection of pro-B cells, large pre-B cells and small pre-B cells. (B) Diagram showing that some c-kit+/CD43+/IgM- bone marrow B-cells overlap with CD25-/IL-7R+/IgM-/CD24+/CD43lo gate for large pre-B cells. (C) Frequency of individual VH in VDJseq libraries from biological replicate libraries from mouse bone marrow pools. Consult Appendix Table S1 for composition of bone marrow B-cell pools. (D) VH frequency in wild-type pro-B non-productive VDJ versus μMT pro-B non-productive VDJ. Data Information: Data derived from RStudio analysis of merged biological replicate datasets of VDJseq analysis of respective cell types, as Fig. 1. $R^2$ values calculated using simple linear regression. Source data are available online for this figure.

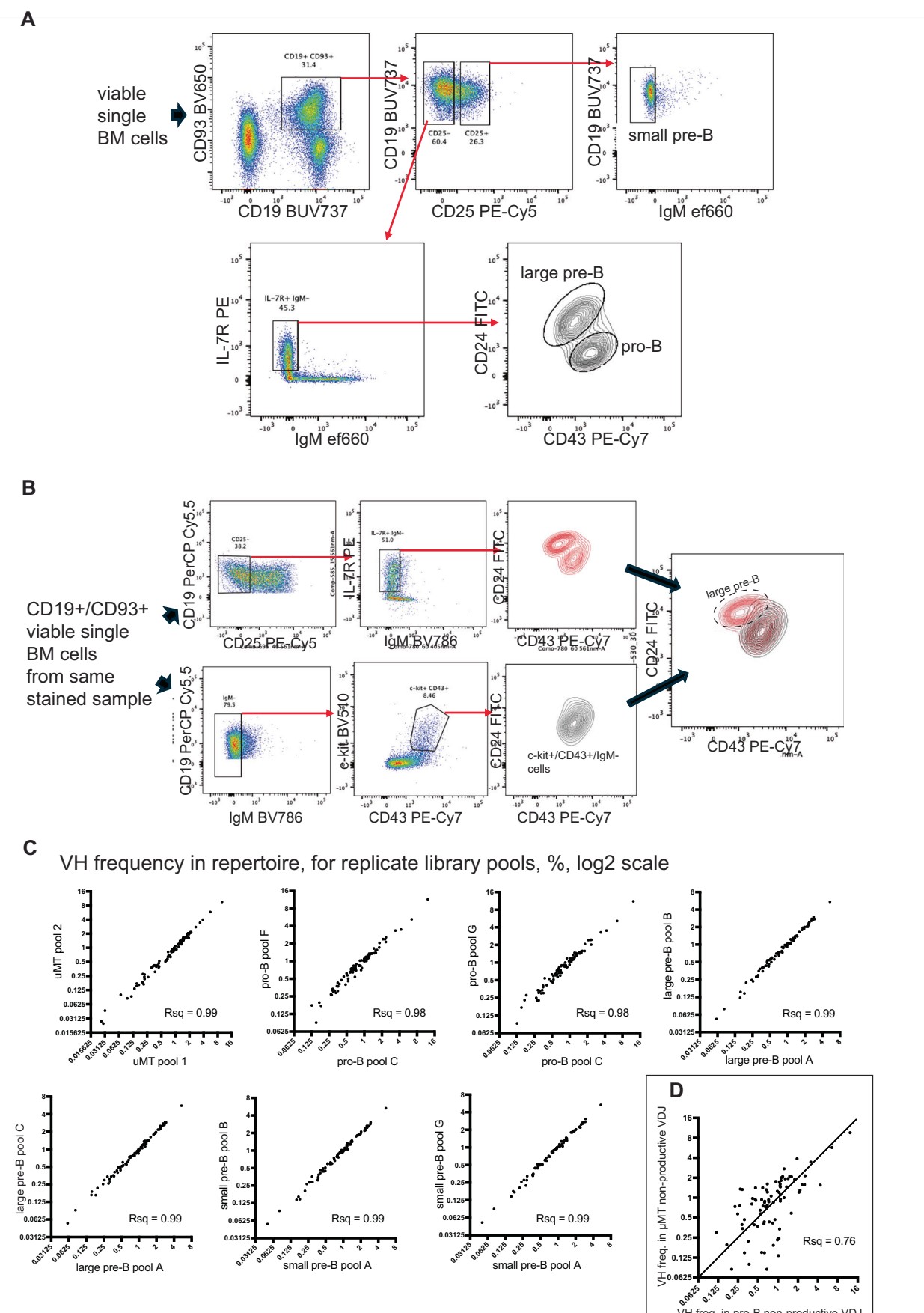

VH frequency in repertoire, for replicate library pools, %, log2 scale

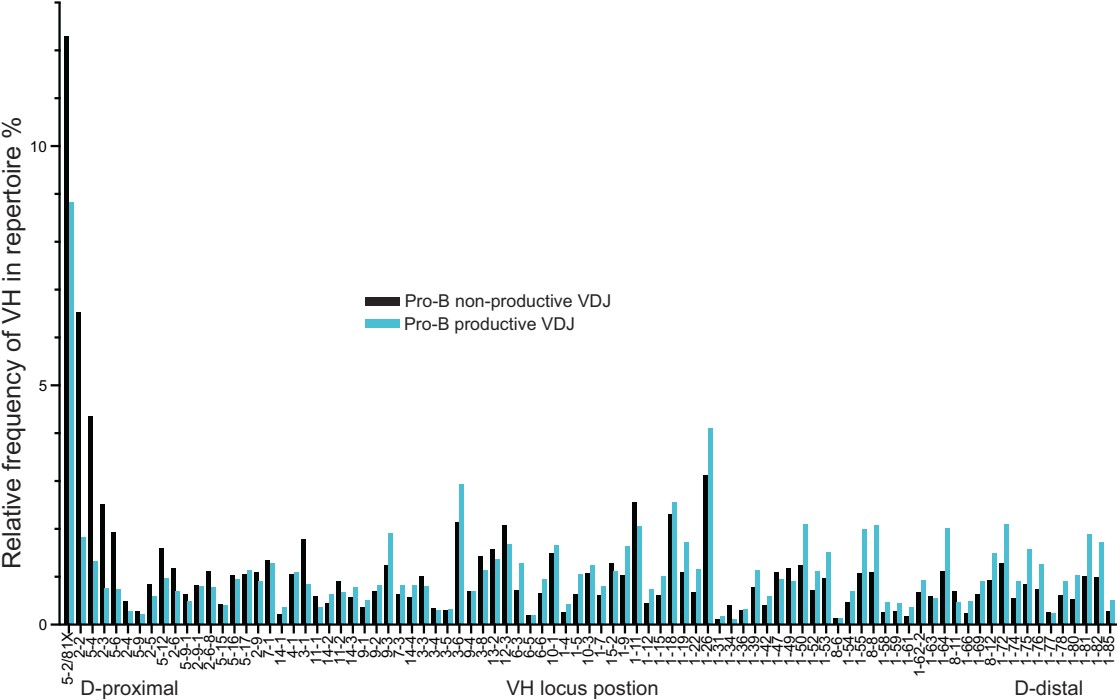

**Figure EV2. VH selection in pro-B cells by locus position.**

VH frequency in non-productive and productive VDJ from pro-B cells, by locus position, indicating the selection of productive VDJ that occurs prior to pre-BCR driven proliferation and differentiation into small pre-B cells. Data Information: Data derived from RStudio analysis of merged biological replicate datasets of VDJseq analysis of respective cell types, as Fig. 1. Source data are available online for this figure.

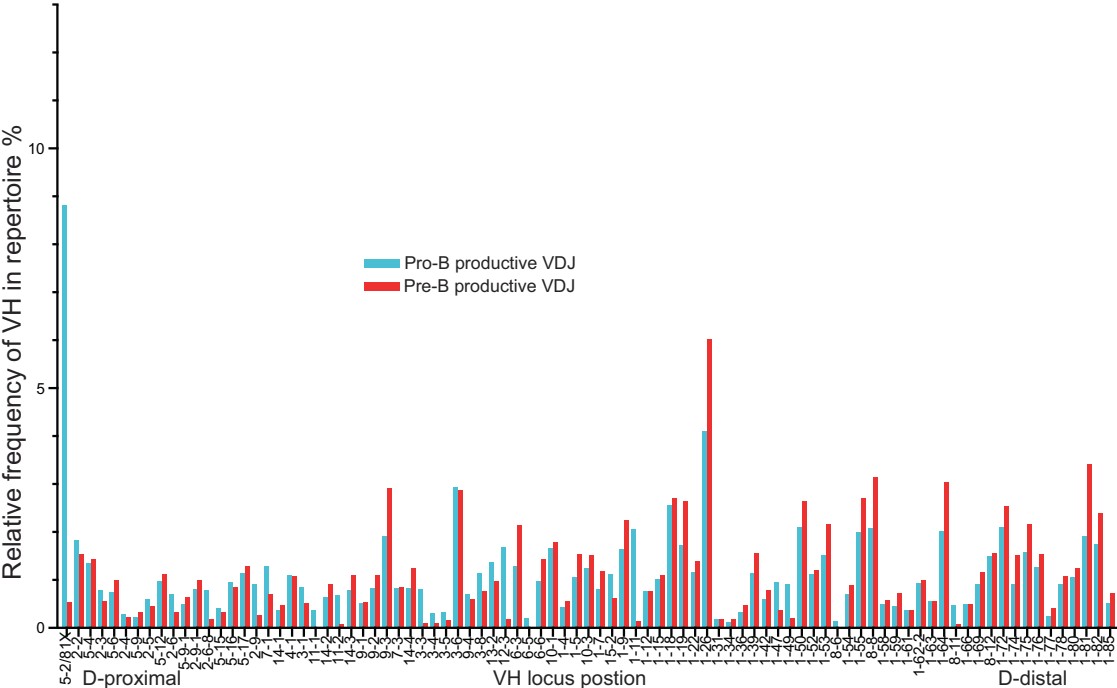

**Figure EV3.   VH selection after the pre-B transition.**

VH frequency in productive VDJ from pro-B cells and pre-B cells, by locus position, indicating the further selection occurring to productive VDJ after completion of the pre-B transition. This selection is in addition to that shown in Figure EV2, together accounting for the overall selection shown in Fig. 2A. Data Information: Data derived from RStudio analysis of merged biological replicate datasets of VDJseq analysis of respective cell types, as Fig. 1. Source data are available online for this figure.

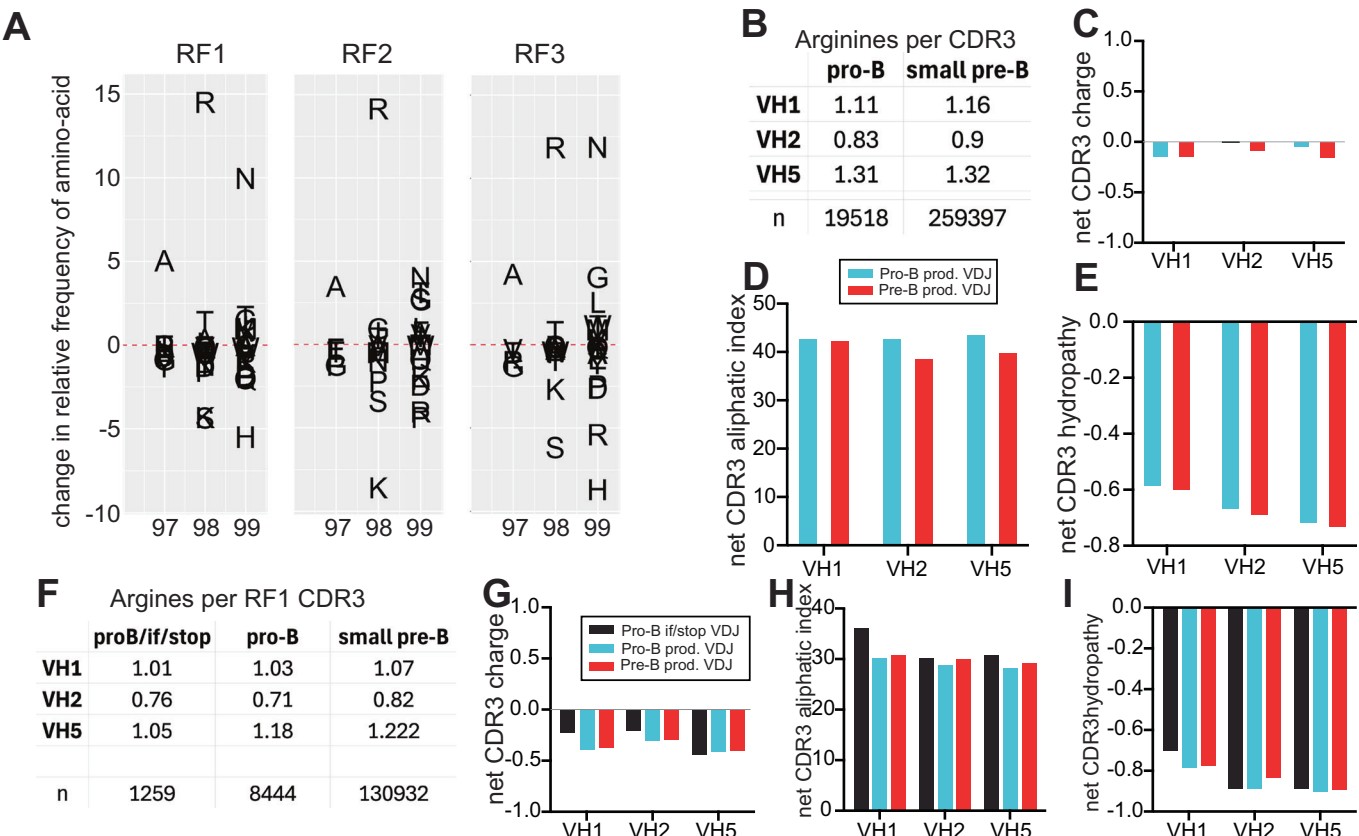

**Figure EV4. Consistent selection of N-terminal CDR3 amino acids in all D reading frames, and biophysical properties of CDR3s, over the pre-B transition.**

(A) Change in frequency of the first three CDR3 amino acid residues over the pre-B transition by D-reading-frame. Data shown here for VH2 only which also shows the strongest Y101 selection and RF2 counter-selection. (B) Mean number of Arginine residues per CDR3 for VH1, VH2, VH5 families from pro-B productive VDJ and pre-B productive VDJ. (C) Net CDR3 charge for VH1, VH2, VH5 families from pro-B productive VDJ and pre-B productive VDJ. (D) Net CDR3 aliphatic index for VH1, VH2, VH5 families from pro-B productive VDJ and pre-B productive VDJ. (E) Net CDR3 hydropathy for VH1, VH2, VH5 families from pro-B productive VDJ and pre-B productive VDJ. (F) Mean number of Arginine residues per RF1 CDR3 for VH1, VH2, VH5 families from pro-B non-productive (in-frame but with a stop codon), pro-B productive VDJ and pre-B productive VDJ. (G) Net RF1 CDR3 charge for VH1, VH2, VH5 families from pro-B non-productive (in-frame but with a stop codon), pro-B productive VDJ and pre-B productive VDJ. (H) Net RF1 CDR3 aliphatic index for VH1, VH2, VH5 families from pro-B non-productive (in-frame but with a stop codon), pro-B productive VDJ and pre-B productive VDJ. (I) Net RF1 CDR3 hydropathy for VH1, VH2, VH5 families from pro-B non-productive (in-frame but with a stop codon), pro-B productive VDJ and pre-B productive VDJ. Data Information: Panels A/B/F data derived from RStudio analysis of merged biological replicate datasets of VDJseq analysis of respective cell types, as Fig. 1. For panels C/D/E/G/H/I data was then further analysed using the method and algorithm from Protparam, https://web.expasy.org/protparam. Source data are available online for this figure.

