## [Peer Review File · The EMBO Journal]

VH-replacement shapes the antibody repertoire by removing the genes of non-functional heavy-chains

Harry White, Peter Chovanec, Laura Biggins, Elise French, Georgia Bullen, Simon Andrews, and Anne Corcoran

Corresponding author(s): Harry White (harry.white@babraham.ac.uk)

Review Timeline:

Submission Date:	9th Jan 25
Editorial Decision:	13th Mar 25
Revision Received:	5th Jun 25
Editorial Decision:	30th Jun 25
Revision Received:	9th Jul 25
Accepted:	25th Jul 25

Editor: Ioannis Papaioannou

Transaction Report:

Dear Dr. White,

Thank you for submitting your manuscript EMBOJ-2025-120103 for consideration by The EMBO Journal, and for your patience during peer review. Your manuscript has now been seen by three experts in the field, and we have received the full set of their comments, which you can find below.

As you will see, the referees are broadly supportive of the work and the manuscript recognizing that the topic is important, the manuscript presents a large amount of appropriately analyzed data, and the overall advance over the literature is significant. They also identify a number of limitations, however, and they provide detailed and reasonable suggestions for strengthening the manuscript further, which we think would increase the impact of the work on the field.

In the context of some of the referees' comments, I would like to point out that The EMBO Journal has a very broad readership of molecular biologists working in all major fields of biology, and it is therefore important that the data are presented in the most accessible manner possible by our broad audience, beyond the field specialists.

Given the referees' positive comments and recommendations, I would like to invite you to submit a revised version of the manuscript along with a detailed point-by-point response addressing all referees' comments and taking the comment above on board. I should add that it is The EMBO Journal policy to allow only a single round of major revision, and acceptance of your manuscript will therefore depend on the completeness of your responses in this revised version. Please let me know if you have any questions or comments that you would like to discuss with me.

We generally allow three months as standard revision time (June 12, 2025). As a matter of policy, competing manuscripts published during this period will not negatively impact our assessment of the conceptual advance presented by your study. However, we request that you contact us as soon as possible upon publication of any related work, to discuss how to proceed. Should you foresee a problem in meeting this three-month deadline, please let us know in advance and we may be able to grant an extension.

Thank you for the opportunity to consider your work for publication in The EMBO Journal. I look forward to your revision.

Best regards,

Ioannis

Instructions for preparing your revised manuscript

1. When you are ready to submit the revision, please upload:

- A Word file of the manuscript text (including legends of main Figures, EV Figures and Tables). Please make sure that changes are highlighted (or "tracked") to be clearly visible.

- Individual production-quality figure files (one file per figure). When assembling your figures, please refer to our figure preparation guidelines in order to ensure proper formatting and readability in print as well as on screen:

If the data shown in a figure are obtained from n {less than or equal to} 2, please use scatter plots showing the individual data points.

- i. the name of the statistical test used to generate error bars and P values
- ii. the number (n) of independent experiments (please specify technical or biological replicates) underlying each data point (discussion of statistical methodology can be reported in the Materials and Methods section, but figure legends should contain a basic description of n , P , and the test applied)
- iii. the nature of the bars and error bars (s.d., s.e.m.).

- A point-by-point response to the referees' comments, with a detailed description of the changes made (as a word file). All referees' concerns must be fully addressed and their suggestions taken on board. When preparing your letter of response to the referees' comments, please bear in mind that this will form part of the Review Process File and will therefore be available online to the community. Please note that you have the possibility to opt out of the transparent process at any stage prior to publication by letting the editorial office know (contact@embojournal.org); if you do opt out, the Review Process File link will point to the following statement: "No Review Process File is available with this article, as the authors have chosen not to make the review process public in this case.". For more details on our Transparent Editorial Process, please visit our website: <https://www.embopress.org/page/journal/14602075/authorguide#transparentprocess>

- Expanded View (EV) files (replacing Supplementary Information) that are collapsible/expandable online. A maximum of 5 EV Figures can be typeset. EV Figures should be cited as "Figure EV1, Figure EV2" etc. in the text, and their respective legends should be included in the manuscript file after the legends of regular figures. See detailed instructions regarding Expanded View files here: <https://www.embopress.org/page/journal/14602075/authorguide#expandedview>

- For the figures that you do NOT wish to display as Expanded View figures, they should be bundled together with their legends in a single PDF file called "Appendix", which should start with a short Table of Contents (including page numbers). Appendix figures should be referred to in the main text as: "Appendix Figure S1, Appendix Figure S2" etc. Please see detailed instructions here: <https://www.embopress.org/page/journal/14602075/authorguide#expandedview>

- A complete author checklist, which you can download from our author guidelines (<https://www.embopress.org/page/journal/14602075/authorguide>). Please note that the checklist will also be part of the Review Process File.

2. Please note that no statistics should be calculated and shown in Figures if $n=2$. Please also note that each p value should be reported as an exact value.

3. Before submitting your revision, primary datasets (and computer code, where appropriate) produced in this study need to be deposited in appropriate public databases (see <https://www.embopress.org/page/journal/14602075/authorguide#dataavailability>). In particular, we kindly request you to deposit all sequencing data produced in the study. The accession numbers, database, and the specific URLs (links) should be listed in a formal "Data availability" section (placed after Methods), following the example below:

"The RNA-seq datasets produced in this study are available in the following database:
Gene Expression Omnibus GSE46843 (<https://www.ncbi.nlm.nih.gov/geo/query/acc.cgi?acc=GSE46843>)"

*** All links should resolve to a page where the data can be accessed. ***

*** Please remember to provide in the Data availability section of your revised manuscript reviewer passwords if the datasets are not yet public. ***

*** The Data Availability Section is restricted to new primary data that are part of this study. In case you have no data that require deposition in a public database, please state so instead of referring to the database: "Our study includes no data deposited in public repositories." under the heading "Data availability". ***

4. The materials and methods need to be described in the manuscript using our structured methods format, which is now required for all research articles. According to this format, the Methods section includes a single "Reagents and Tools Table" - listing key reagents, experimental models, software and relevant equipment including their sources and relevant identifiers - followed by a "Methods and Protocols" section describing the methods. Please download and fill our Reagents and Tools Table template (.docx), which you can find in our author guide:

<https://www.embopress.org/page/journal/14602075/authorguide#structuredmethods>. When submitting your revised manuscript, please do not include the Reagents and Tools Table in the Methods section of the manuscript but instead upload it as a separate file choosing the file type "Reagent Table".

5. Please check that the title and the abstract of the manuscript are brief, yet explicit, even to non-specialists. The length of the title should not exceed 100 characters, and the abstract should be a single paragraph not exceeding 175 words.

6. Please also note our reference format: <https://www.embopress.org/page/journal/14602075/authorguide#referencesformat>.

8. Please remember: digital image enhancement is acceptable practice, as long as it accurately represents the original data and conforms to community standards. If a figure has been subjected to significant electronic manipulation, this must be noted in the figure legend or in the "Materials and Methods" section. The editors reserve the right to request original versions of figures and the original images that were used to assemble the figure.

9. Our journal encourages inclusion of data citations in the reference list to directly cite datasets that were obtained from public databases. Data citations in the article text are distinct from normal bibliographical citations and should directly link to the database records from which the data can be accessed. In the main text, data citations are formatted as follows: "Data ref: Smith et al, 2001" or "Data ref: NCBI Sequence Read Archive PRJNA342805, 2017". In the Reference list, data citations must be labeled with "[DATASET]". A data reference must provide the database name, accession number/identifiers, and a resolvable link to the landing page from which the data can be accessed at the end of the reference. Further instructions are available at: <https://www.embopress.org/page/journal/14602075/authorguide#referencesformat>.

10. We request authors to consider both actual and perceived competing interests. Please review our policy (<https://www.embopress.org/page/journal/14602075/authorguide#conflictsofinterest>) and update your competing interests statement if necessary. Please name this section 'Disclosure and competing interests statement' and place it after the Acknowledgements section.

11. Please note that all corresponding authors are required to provide an ORCID ID upon submission of a revised manuscript (<https://orcid.org/>). Please find instructions on how to link your ORCID ID to your account in our manuscript tracking system in our Author guidelines (<https://www.embopress.org/page/journal/14602075/authorguide#authorshipguidelines>).

12. We use CRediT to specify the contributions of each author in the journal submission system. CRediT replaces the author contribution section, which should be removed from the manuscript. Please use the free text box to provide more detailed descriptions. See also guide to authors: <https://www.embopress.org/page/journal/14602075/authorguide#authorshipguidelines>.

14. We would also welcome the submission of cover suggestions or motifs to be used by our Graphics Illustrator in designing a cover.

15. Please use the link below to submit your revision:
<https://emboj.msubmit.net/cgi-bin/main.plex>

Referee #1:

This manuscript examines Vh repertoire changes and Vh replacement during bone marrow B cell development. These are important and interesting issues and the data (almost entirely repertoire sequence data) are extensive and thoughtfully analyzed. Comparison of repertoires from μ MT and WT mice add significantly to the analyses. The data provide numerous detailed insights into the selective forces acting on individual and groups of Vh gene segments and on amino acid residues at particular positions in CDR3. The most significant conclusion of the paper relates to evidence showing that Vh replacement acts at the pro-B stage to replace productively rearranged Vh segments that are unable to pair efficiently with the surrogate light chains. The raison d'être of Vh replacement has not been completely resolved and this work helps fill that gap. Overall this is an excellent study that advances our understanding of the forces that shape Vh repertoire significantly.

There are some ways the manuscript could be improved. Major suggestions:

1. Lines 404-406, the authors make the following important assertion. " μ MT mice do not undergo the VDJ selection (Figure 4a) that we show is impacted by VHR, confirming that this selection is driven by μ -chain signalling and thus productive VDJ expression." If this is true, then Vh replacement in μ MT pro-B cells should be rare. This is readily testable using the PCR assay of Fig. 5a. I suggest that the authors perform the assay on μ MT pro-B cell gDNA, focused on V5-2/81X, to determine whether their assertion is true. This is important and would strengthen the manuscript because it relates directly to the central conclusion of the paper that Vh replacement functions to replace functional but non-pairing VDJ rearrangements.
2. Many nice plots are provided showing pair-wise comparisons of Vh usage frequencies (productive and non-productive at different stages of development in WT and μ MT mice). However, there are no plots that allow readers to see all of the data together. To do so for all Vh segments would be too complicated and too much. I wonder if it would be possible, for selected and

informative Vh segments, to provide summary plots bringing all of the frequency data together.

More minor points:

3. Line 19, abstract, and line 116: the claim that this is the "first" such analysis should be removed. This claim is vague and hard to verify. While this might be the most in-depth study, this topic has been addressed previously. It would suffice to say "undertaken an in-depth analysis".
4. Line 74: CDR as an abbreviation has not been defined, here or in the abstract.
5. There are places where μ MT (with a Greek letter) is written mMT, e.g., legend to Figure 1. Please correct.
6. At the beginning of the results, it would be helpful to readers to devote a sentence or two to describe the repertoire sequencing method utilized. And, it would be appropriate to address any limitations associated with the method. After all, this method is the source of all of the data for the paper.
7. The Vh nomenclature is challenging to follow. For example, in Fig. 2, Vh segments are grouped three different ways (classic family, family, and clan). The table in c helps somewhat, but it would be very nice if the data plotted in b could be visualized/understood by family rather than classic family, since it is family that is referred to most often in the text. Perhaps some sort of color coding?
8. Paragraph from lines 283-293: please tell the reader where to look in the figures for the data backing up the statements made.
9. Statement on lines 314-315 does not seem to effectively describe the data of Fig. 4d in two regards. First, the association for clan 3 is unconvincing, since even those with no cryptic RSS show low productivity per Vh. And second, the claim that clan 1 Vh as a group demonstrates a correlation is unconvincing, since a large proportion of those with a cryptic RSS show high productivity per Vh (the type of statistical test used is not specified in the figure legend). It is not clear that claims based on entire clans (for clans 1 and 3) are justified here. Rather, there appears to be specific gene segments that show the association being discussed and are outliers from the rest of the clan.
10. While the Igh locus is depicted from left to right in schematics (e.g., Fig. 5a) (and this is the conventional orientation used in most depictions of antigen receptor loci and is the most intuitive for those who read left to right), all of the locus data are depicted with the locus oriented from right to left. This is disorienting and at times confusing. It would likely be a huge amount of work to recreate all of the figures in the other orientation, and I'm not requiring it, but I would like to understand why the authors use this orientation.

Referee #2:

White et al. EMBO J VH-replacement (VHR)

Overall, this is a valuable contribution, both in terms of providing a resource for the field and by demonstrating a major impact of VH gene replacement in the development of the antibody repertoire in the mouse, under physiological conditions. Unfortunately, the manuscript is exceedingly difficult to read, contains overstatements, and often lacks clarity, so that extensive editing is required before publication. Below I provide an incomplete list of a number of criticisms and suggestions, not ordered according to importance:

Fig. 1e: Productive (legend) or non-productive (y-axis) VDJ?

Fig. 4a-d: What does the labeling of the ordinates ("productivity per VH/100%") mean? Fraction of productive VDJ segments?

Lines 17-18: "Critically,..." and lines 446-452: "The bone marrow cytokine milieu...": Eliminate, because vague and unrelated to the content of the paper.

Line 292: "non-pairing" on what basis? Delete or explain.

Line 316: "non-pairing" on what basis? Delete or explain.

Lines 384-387: "...resolves a long-lasting question fundamental to VDJ repertoire formation, that of what happens to the non-pairing VDJ" seems overstated to this referee. Could the authors put in precise statements on what they really resolve? Thus: What exactly allows them to conclude, or do they even want to conclude, that VHR works more frequently on productive, non-pairing VDJs than on

pairing or non-productive ones? That VHR can efficiently modify productive VDJs was known before, from work in transgenic mouse models, but also in the human, in a cell line as well as primary B lineage cells (Zhang Z et al. *Immunity* 2003; also describing sequential VHR, not addressed in the present work), accompanied by selection of charged amino acid residues in CDR3 etc. The Zhang et al work is not cited/discussed in the present manuscript - it should be! Line 392: Is the "only" meant to refer to productive rearrangements only? Please clarify.

Line 406: Delete "productive".

Line 409: Delete "productive".

Line 414: Introduce comma following "pre-BCR".

Lines 424-426: "This suggests that VHR can account for all the changes in productive VDJ we observe, considering the likely extended pro-B cell phase of a non-pairing productive VDJ in a normal physiological context." The meaning and logic of this sentence remains obscure. To what exactly does the "This" in the beginning of the sentence refer, and how does the statement in the first half of the sentence then connect to "considering the likely extended pro-B cell phase of a non-pairing productive VDJ in a normal physiological context"? And, more basically, how can the sweeping claim in the first half sentence be made in view of the well-established (and VHR independent) principle of pre-BCR driven proliferative expansion preceding the "pre-B cell transition"?

Referee #3:

The work offers an interesting hypothesis regarding the functional utility of a type of secondary RAG-mediated gene rearrangement. This process was carefully studied a decade or more ago but the authors' use of high throughput genetic methods reveal an interesting novelty. I am unable to judge whether that particular insight - an issue of lymphocyte antigen-receptor clonal specificity - is appropriate for much of the *EMBO J*'s readership. The manuscript of White et al. evinces in this reviewer that feeling between old lovers who meet after a decade of separation. Each knows the other intimately but there remain new things to discuss.

White et al. describe new work on the selection of favored VDJ rearrangements as pro-B cells generate functional Mu chain able to form the pre-BCR by productive interaction with the surrogate light-chain (SLC). This critical step in B-cell development has long been known to shape the repertoire of antigen receptors available to mature B-cell pools; the efficacy of Mu:SLC interaction is well known as the critical determinant for producing a functional pre-BCR that sustains maturation through pre-B cell differentiation.

Likewise, numerous studies have demonstrated "receptor editing", secondary Vh into VDJ rearrangements in pro-B cells at a widely conserved cryptic RSS (cRSS) frequently located at the 3' border of the Vh gene segment. This receptor "editing event" generates a novel, chimeric Mu polypeptide altering the potential of the resultant pre-BCR and BCR. Significantly, functional cRSS are also conserved in mouse Vh genes at sites unsuitable for this sort of editing event (cf the listed Cowell and Davila references) - more later.

The authors state (L116), "We undertook the first in-depth analysis of VDJ selection over the pre-B transition" but mean, I think, the first analysis using contemporary DNA sequencing methods and analysis (L534) "pipelines". This work is nicely done and corroborates and occasionally demonstrates much prior work on this topic. The advantage of the new work is numbers, primarily. And that in no way diminishes the significance of the study. The authors show convincingly that productive VDJ recombination to generate functional Mu protein is inefficient - only about 25% - due to reading frame bias and other intrinsic limitations of the recombination process. That was known of course but the accounting possible in this study is novel. Selection for Mu across the pro-B to pre-B transition is observed, most notably the Vh5-2 (V81X) segment, confirming prior work from the Alt laboratory, but now on the population level. Much of this selection is linked to Jh bias and a presumed disfavored interaction with SLC and the Mu 3rd hypervariable region (HVR).

The first key utility of the population numbers available for this study is the finding of specific 3rd HVR residues with pre-BCR selection (LL 193-238). These findings are novel and implicate a critical role for this site in favored/disfavored interaction with the SLC. One can easily imagine experiments *in vivo* to test the authors' hypotheses.

The second is the authors' observation that certain early/proximal Vh gene segments appear to be major targets for receptor editing events, as inferred by their absence as productive but non-functional Mu rearrangements. The authors' genetic

population surveys show that the "missing" fraction of in-frame rearrangements of early/proximal Vh gene segments can be explained plausibly by favored Vh gene replacement. The dynamite point here is that the Vh replacement is riding the pro-B cell of a second, in-frame Mu protein that does not interact with the SLC. Vh replacement acts as a mechanism to ensure single BCR specificity. This is very nice work and again a testable hypothesis (I am not impressed by the in vitro culture experiment shown in Fig. 3 given heterogeneity in the actual differentiative stages of the collected pro-B cells). To that end, Cowell and Davila references point out the surprising result that conserved cRSS can be found scattered about Vh gene segments, not only in the 3' site associated with the traditional receptor editing observations by Wiegert and others. Obviously, if the physiological role of cRSS in Vh genes is inactivation, their location doesn't really matter.

Response to Reviewers:

We sincerely thank all three reviewers for their extensive, thorough and constructive comments. Responding to these has improved the manuscript.

Below are our responses to the specific comments

All our responses to referees comments below are in Blue text

Referee 1

This manuscript examines Vh repertoire changes and Vh replacement during bone marrow B cell development. These are important and interesting issues and the data (almost entirely repertoire sequence data) are extensive and thoughtfully analyzed. Comparison of repertoires from μ MT and WT mice add significantly to the analyses. The data provide numerous detailed insights into the selective forces acting on individual and groups of Vh gene segments and on amino acid residues at particular positions in CDR3. The most significant conclusion of the paper relates to evidence showing that Vh replacement acts at the pro-B stage to replace productively rearranged Vh segments that are unable to pair efficiently with the surrogate light chains. The raison d'être of Vh replacement has not been completely resolved and this work helps fill that gap. Overall this is an excellent study that advances our understanding of the forces that shape Vh repertoire significantly.

There are some ways the manuscript could be improved. Major suggestions:

1. Lines 404-406, the authors make the following important assertion. " μ MT mice do not undergo the VDJ selection (Figure 4a) that we show is impacted by VHR, confirming that this selection is driven by μ -chain signalling and thus productive VDJ expression." If this is true, then Vh replacement in μ MT pro-B cells should be rare. This is readily testable using the PCR assay of Fig. 5a. I suggest that the authors perform the assay on μ MT pro-B cell gDNA, focused on V5-2/81X, to determine whether their assertion is true. This is important and would strengthen the manuscript because it relates directly to the central conclusion of the paper that Vh replacement functions to replace functional but non-pairing VDJ rearrangements.

We thank the referee for this important and insightful suggestion. We have done this experiment and find a 2.5 fold reduction in VH-replacement circles in equivalent amounts of μ MT gDNA. (new Fig 5E)

This experiment has caveats which we have described as follows:

Additional text in Results:

Line 135:

Due to minor differences we detect between the C57BL/6 and μ MT, described in the methods section, we are not directly comparing VH repertoire frequencies between these strains, but using μ MT mice to analyse other fundamentals of VDJ recombination.

Line 676:

If VH-replacement is driving the removal of productive non-pairing VDJ in wild-type pro-B cells, explaining the early VDJ selection we observe, and this selection is largely absent in μ MT pro-B cells (Figure 4A), it follows that such VH-replacement is enhanced by μ -chain signalling. If this is the case, we should find fewer VH-replacement circles in μ MT pro-B cells. Due to differences in μ MT pro-B cell VDJ, described in the methods section, we could only search for replacement-circles using the shell-script search for back-to-back RSS. In our hands this detects around half of the circles we would find by using the more sophisticated search, described in methods, in addition. We analysed RCseq libraries made from identical amounts of DNA (540ng) from one wild-type pro-B cell pool and two μ MT pro-B cell pools. The results are shown in figure 5E. We found five VH-replacement circles in the wild-type pro-B-cell sample and only two in each of the μ MT pro-B cell samples. The replacement-circle sequences are shown in Appendix Table E. This result supports the proposal that μ -chain signalling enhances VH-replacement.

Additional text in Discussion:

Line 753:

If this is the case, we should find fewer VH-replacement circles in μ MT mouse pro-B-cells, and Figure 5E shows data supporting this hypothesis. We found a 2.5-fold higher frequency of VHR circles from a pool of WT mouse pro-B cells as compared to two pools of μ MT mouse pro-B-cells. Of additional note is that 4/5 of the replacement circles in WT mice showed VH5-2/81X replacement whereas 0/4 of the replacement circles from μ MT mice contained VH5-2/81X (Appendix Table E).

From Methods:

Line 1011: Use of μ MT pro-B cells

Despite extensive back-crossing to C57BL/6 mice, the 129-mouse derived μ MT *Igh* locus is likely slightly different from that of C57BL/6. Nevertheless, VDJseq reproducibly detects 88 of the 89 VH we analyse, in biological replicates of μ MT pro-B cell pools (Figure EV1C, $R^2 = 0.99$). Whilst there is also strong correlation in VH frequency between C57BL/6 and μ MT non-productive VDJs, however, there is still some variation (Figure EV1D, $R^2 = 0.76$). For this reason we are not directly comparing VH frequencies between these strains, but using μ MT mice to analyse other fundamentals of VDJ recombination

2. Many nice plots are provided showing pair-wise comparisons of Vh usage frequencies (productive and non-productive at different stages of development in WT and μ MT mice). However, there are no plots that allow readers to see all of the data together. To do so for all Vh segments would be too complicated and too much. I wonder if it would be possible, for selected and informative Vh segments, to provide summary plots bringing all of the frequency data together.

We have considered at length in the past how to integrate this type of data for more summary presentation. Some studies (e.g. Kaplinsky et al. PNAS 2014) have used principal component analysis, but unlike for gene expression studies where it is helpful simply to cluster cell types, we consider the lack of biological meaning to component dimensions renders this approach less helpful for repertoire analysis.

To address the comment, however, we have amended two sets of summary data which we hope deal with some of the issues. In particular, we have put old Figure 2a in figure 1 adjacent to another similar figure, and included a 3rd new plot. These have become Figures 1F-H. These show the VH frequencies in VDJ from comparison datasets as scatter plots and give a good overall picture of the selection or lack of it, between μ MT non-productive vs productive; WT pro-B non-productive vs productive, and WT pro-B productive vs pre-B productive. The last two plots clearly show the selection occurring before and after pre-B cell proliferation. In addition, we have linked these last two figures through the 'Extended View' feature to their cognate histograms which show the individual VH frequencies by locus position. We think this will greatly enhance the ease of assessing this rather dense data.

The associated text changes are from line 227.

We have also slightly amended Figure 5D. This now includes VH5-2/81X and shows a selection of VH that do and don't lose productive VDJ over the pre-B transition, for reference to those VH that are subject to VH-replacement

More minor points:

3. Line 19, abstract, and line 116: the claim that this is the "first" such analysis should be removed. This claim is vague and hard to verify. While this might be the most in-depth study, this topic has been addressed previously. It would suffice to say "undertaken an in-depth analysis". We thank the reviewer for this comment and have made the necessary alteration

4. Line 74: CDR as an abbreviation has not been defined, here or in the abstract. Thank you for pointing this out, we have defined CDR in the Abstract. Line 27: complementarity-determining-region 3 (CDR3).

5. There are places where μ MT (with a Greek letter) is written mMT, e.g., legend to Figure 1. Please correct.

Thank you for drawing our attention to this inconsistency. We have corrected this issue

6. At the beginning of the results, it would be helpful to readers to devote a sentence or two to describe the repertoire sequencing method utilized. And, it would be appropriate to address any limitations associated with the method. After all, this method is the source of all of the data for the paper.

We thank the reviewer for this comment and have included a new paragraph describing the method and its limitations. This has also given us the opportunity to more formally define the nomenclature and sorting of the B-progenitors, which has also improved the manuscript. The paragraph is as follows:

Line 113: Progenitor B-cell definition and immunoglobulin heavy-chain repertoire analysis

We undertook an in-depth analysis of VDJ selection over the pre-B transition, focusing on the VH repertoire of pro-B cells (Basle 'pre-BI') and small pre-B cells (Basle 'small pre-BII'), sorted according to the scheme in Figure EV1A. This adapts the use of markers defined by Hardy (Hardy et al., 1991) and the Basel group (Rolink & Melchers, 1996). We find this approach more stringently defines pro-B cells. In our hands the conventional FACS sorted CD43+/IgM-/c-kit+ pro-B cell phenotype includes a subset of cells with a large pre-B cell phenotype (Figure EV1B). B-cell genomic DNA was then subject to quantitative immunoglobulin VDJ repertoire analysis using VDJseq (Bolland et al., 2016; Chovanec et al., 2018). This next generation sequencing technique captures J-segment containing sequences and detects both DJ and VDJ recombinants, allowing measurement of the frequency of VDJ recombinants in populations. The VDJseq approach is highly reproducible, showing strong correlation for biological replicate samples (Figure EV1C). Whilst this method may be limited by a modest sequence sampling rate of 5-10% of sequences in the starting gDNA pool, (Chovanec et al., 2018), this was overcome with use of bone-marrow from larger pools of mice.

7. The Vh nomenclature is challenging to follow. For example, in Fig. 2, Vh segments are grouped three different ways (classic family, family, and clan). The table in c helps somewhat, but it would be very nice if the data plotted in b could be visualized/understood by family rather than classic family, since it is family that is referred to most often in the text. Perhaps some sort of color coding?

We thank the reviewer for highlighting this lack of clarity. We have clarified the figures by removing references to classic VH-families.

8. Paragraph from lines 283-293: please tell the reader where to look in the figures for the data backing up the statements made.

We thank the reviewer for pointing out this omission and have fixed it. Line 530 onwards

9. Statement on lines 314-315 does not seem to effectively describe the data of Fig. 4d in two regards. First, the association for clan 3 is unconvincing, since even those with no cryptic RSS show low productivity per Vh. And second, the claim that clan 1 Vh as a group demonstrates a correlation is unconvincing, since a large proportion of those with a cryptic RSS show high productivity per Vh (the type of statistical test used is not specified in the figure legend). It is not clear that claims based on entire clans (for clans 1 and 3) are justified here. Rather, there appears to be specific gene segments that show the association being discussed and are outliers from the rest of the clan.

We thank the referee for this helpful comment. We agree, after reflection, that interpreting the data in Figure 4D according to clan is not robust. Accordingly we have adjusted the text.

During further analysis of the VH without cRSS that generate the lowest productivity VDJ, to further address the referees comments, we noticed that the two VH that generate the lowest productivity VDJ, end in the first two base-pairs of a stop codon suggesting they are prone to generating non-productive VDJ. Only one other VH has this feature and it also has low productivity. Addressing the comments has clarified and improved this data and we have amended the whole paragraph as follows:

From line 576: Cryptic-RSSs facilitate VH-replacement, so we analysed their association with VDJ productivity. Low VDJ productivity correlates strongly with the presence of a cRSS in the VH (Figure 4D). This picture is complicated, however, since many VH with a cRSS have higher productivity VDJ. This suggests possession of a cRSS is necessary but not sufficient to render a VDJ vulnerable to VH-replacement, and other factors are important for example the frequency of formation of non-pairing VDJ. There are a few VH without cRSS that generate low productivity VDJ (Figure 4D). It is notable, however, that the two VH without cRSS with the lowest productivity rates (VH11-1/11-2), marked with blue dots, have V-segments that end with the first two base-pairs of a stop codon (TA). This suggests these VH, along with the single other VH ending TA (VH3-8, indicated in Figure 4D), have an intrinsic tendency to form non-productive VDJ. Overall, these results are consistent with VH replacement of certain productive VDJ in pro-B cells, mediated by recombination at the cRSS.

10. While the Igh locus is depicted from left to right in schematics (e.g., Fig. 5a) (and this is the conventional orientation used in most depictions of antigen receptor loci and is

the most intuitive for those who read left to right), all of the locus data are depicted with the locus oriented from right to left. This is disorienting and at times confusing. It would likely be a huge amount of work to recreate all of the figures in the other orientation, and I'm not requiring it, but I would like to understand why the authors use this orientation.

We agree that it is more common in the literature to plot diagrams of the *Igh* locus from 5' to 3', left to right. Our laboratory has a long history of VH gene identification, sequencing, mapping and epigenetics of recombination, and has largely adhered to the genomics convention of plotting VH genes as they appear on conventional chromosome 12 annotation, i.e. reading C-J-D-V left to right. We have used this convention when developing VDJ-seq (Bolland et al., *Cell Reports* 2016), and more recently in studies of *Igh* chromatin organisation, e.g. Mielczarek et al., *Cell Reports*, 2023. We appreciate this comment and have reversed the schematic in Figure 5A to remove the conflict of conventions, and have relabelled the relevant X axes in Figures 2 to 5 as D-proximal and D-distal for greater clarity.

Referee 2

White et al. EMBO J VH-replacement (VHR)

Overall, this is a valuable contribution, both in terms of providing a resource for the field and by demonstrating a major impact of VH gene replacement in the development of the antibody repertoire in the mouse, under physiological conditions. Unfortunately, the manuscript is exceedingly difficult to read, contains overstatements, and often lacks clarity, so that extensive editing is required before publication. Below I provide an incomplete list of a number of criticisms and suggestions, not ordered according to importance:

Fig. 1e: Productive (legend) or non-productive (y-axis) VDJ?

We thank the referee for noticing this error and have corrected the relevant figure legend which had the mistake

Fig. 4a-d: What does the labeling of the ordinates ("productivity per VH/100%") mean? Fraction of productive VDJ segments?

We thank the referee for highlighting the overcomplicated labelling here. This scale is really just 'Productivity per VH, %', but due to the axes scale being fractional, percentages should be divided by 100. We have re-labelled the Y-axes in the figure to remove the confusion.

Lines 17-18: "Critically,..." and lines 446-452: "The bone marrow cytokine milieu...": Eliminate, because vague and unrelated to the content of the paper.

Regarding old lines 17-18, in an earlier draft of the abstract, this and other since-removed statements, provided some wider context, but after editing, we agree that this statement seems somewhat disembodied. We have removed it as we agree that in its current form it does not relate strongly to the overall content of the paper.

Regarding old lines 446-452, the last paragraph of the discussion, we agree that the wording may be vague, so we have altered it. We choose to leave the amended paragraph in as we consider it accepted practice to comment at the end of a discussion on the wider implications of the work, and further avenues of investigation that may follow from the reported findings. Having determined, in depth, many metrics of heavy-chain selection at the pre-B transition, we consider it well worth instigating further work on how this selection is altered in the startlingly different circumstances of inflammation, where pro-B cells migrate to the spleen and continue development in this altered environment, which also is a feature of many normal immune responses. The amended paragraph is shown here:

Line 869:

B-cell cytokine signalling is altered in ageing and chronically inflamed bone-marrow (Dowery et al., 2021; Koohy et al., 2018; Pioli et al., 2019). Inflammatory stimuli can drive release of B-cell progenitors into the periphery (Nagaoka et al., 2000; Ueda et al., 2004). These stresses significantly impact cells undergoing the pre-B transition, re-locating them to the spleen. This is highly likely to have impacts on the peripheral heavy-chain repertoire. Our in-depth study of the structure and selection of the normal heavy-chain repertoire provides a good foundation to further investigate such alterations.

Line 292: "non-pairing" on what basis? Delete or explain.

We agree with the referees comment that use of the expression 'non-pairing' at this point in the paper is not relevant to the particular data being discussed and have removed it. Line 568

Line 316: "non-pairing" on what basis? Delete or explain.

We agree with the referees comment that use of the expression 'non-pairing' at this point in the paper is not relevant to the particular data being discussed and have removed it, Line 587

Lines 384-387: "...resolves a long-lasting question fundamental to VDJ repertoire formation, that of what happens to the non-pairing VDJ" seems overstated to this referee.

We agree that at this point so early in the discussion this statement could be considered too strong and have changed the word 'resolves', to 'addresses, which also allows further development of arguments in the discussion. Line 711

Could the authors put in precise statements on what they really resolve? Thus: What exactly allows them to conclude, or do they even want to conclude, that VHR works more frequently on productive, non-pairing VDJs than on pairing or non-productive ones?

We thank the referee for highlighting this issue and we have made substantial revision to the discussion to explain our conclusions. The key section now reads as follows:

Line 726: While our strategy cannot determine whether the replaced VDJ was productive or non-productive, VH5-2/81X, VH3-1 and VH1-11, for which we detect VHR, principally demonstrate loss of productive rather than non-productive VDJ over the pre-B transition (Figure 5D). This contrasts with the other VH analysed, that show no loss of productive VDJ and no evidence of VHR.

If productive non-pairing VDJ were functionally equivalent to non-productive VDJ they would show similar drops in frequency over the pre-B transition. The data for VH5-2/81X VDJ, the great majority of which do not pair with the SLC, and which account for half of VHRs we detect, are consistent with VH-replacement with a strong bias toward productive non-pairing VDJ. VH1-11, uniquely, encodes a G97. Supposing this residue interferes with SLC-pairing, we counted the frequency of this residue in non-VH1-11 pro- and pre-B cell productive VDJs, which have presumably appeared through imprecise V-D joins. We found the frequency of G97 in non VH1-11 VDJ drops by around two-thirds in pre-B-cells (from 144/19115 to 719/259018, 0.753% to 0.277%). This suggests that the G97 interferes with SLC-pairing in VH1-11 VDJ making it more likely to be replaced as compared to other VH1 VDJ (e.g. Figure 5C,D).

That VHR can efficiently modify productive VDJs was known before, from work in transgenic mouse models, but also in the human, in a cell line as well as primary B lineage cells (Zhang Z et al. Immunity 2003; also describing sequential VHR, not addressed in the present work), accompanied by selection of charged amino acid residues in CDR3 etc. The Zhang et al work is not cited/discussed in the present manuscript - it should be!

We thank the referee for raising this issue. Our substantial revision of the discussion now includes a more extensive description of what is known about VHR. We have included the reference to Zhang et al. 2003, as well as several others, Line 717

Line 392: Is the "only" meant to refer to productive rearrangements only? Please clarify.

Our revision of the discussion has removed this word and made the sense clearer. Line 727

Line 406: Delete "productive".

Our revision of the discussion has altered this sentence removing the word 'productive'

Line 409: Delete "productive".

We agree, on reflection, that the word 'productive' is redundant if followed by 'non-pairing μ -chain', so have removed it. Line 808

Line 414: Introduce comma following "pre-BCR".

We thank the referee for highlighting this punctuation error. Adding the comma has improved the sense of the sentence.

Lines 424-426: "This suggests that VHR can account for all the changes in productive VDJ we observe, considering the likely extended pro-B cell phase of a non-pairing productive VDJ in a normal physiological context." The meaning and logic of this sentence remains obscure. To what exactly does the "This" in the beginning of the sentence refer, and how does the statement in the first half of the sentence then connect to "considering the likely extended pro-B cell phase of a non-pairing productive VDJ in a normal physiological context"? And, more basically, how can the sweeping claim in the first half sentence be made in view of the well-established (and VHR independent) principle of pre-BCR driven proliferative expansion preceding the "pre-B cell transition"?

We thank the reviewer for this comment. Our original discussion had some lack of clarity and we have altered this particular section to read as follows:

Line 829: That such high levels of VHR can occur, suggests it can account for the changes in productive VDJ levels we observe in pro-B cells prior to pre-BCR mediated proliferation (Figure 1G), especially considering the likely extended pro-B phase of cells with non-pairing μ -chains.

And we have added this sentence to extend the discussion:

Line 832: Also, cells with VHRs that generate pairing μ -chains will rapidly exit the pro-B compartment, suggesting some of the selection seen after pre-B cell proliferation is generated by VHR rather than differential proliferation.

Referee 3

The work offers an interesting hypothesis regarding the functional utility of a type of secondary RAG-mediated gene rearrangement. This process was carefully studied a decade or more ago but the authors' use of high throughput genetic methods reveal an interesting novelty. I am unable to judge whether that particular insight - an issue of lymphocyte antigen-receptor clonal specificity - is appropriate for much of the EMBO J's readership.

The manuscript of White et al. evinces in this reviewer that feeling between old lovers who meet after a decade of separation. Each knows the other intimately but there remain new things to discuss.

White et al. describe new work on the selection of favored VDJ rearrangements as pro-B cells generate functional μ chain able to form the pre-BCR by productive interaction with the surrogate light-chain (SLC). This critical step in B-cell development has long been known to shape the repertoire of antigen receptors available to mature B-cell pools; the efficacy of μ :SLC interaction is well known as the critical determinant for producing a functional pre-BCR that sustains maturation through pre-B cell differentiation.

Likewise, numerous studies have demonstrated "receptor editing", secondary Vh into VDJ rearrangements in pro-B cells at a widely conserved cryptic RSS (cRSS) frequently located at the 3' border of the Vh gene segment. This receptor "editing event" generates a novel, chimeric μ polypeptide altering the potential of the resultant pre-BCR and BCR. Significantly, functional cRSS are also conserved in mouse Vh genes at sites unsuitable for this sort of editing event (cf the listed Cowell and Davila references) - more later.

The authors state (L116), "We undertook the first in-depth analysis of VDJ selection over the pre-B transition" but mean, I think, the first analysis using contemporary DNA sequencing methods and analysis (L534) "pipelines".

We have removed the expression 'the first', Line 114

This work is nicely done and corroborates and occasionally demonstrates much prior work on this topic. The advantage of the new work is numbers, primarily. And that in no way diminishes the significance of the study. The authors show convincingly that productive VDJ recombination to generate functional Mu protein is inefficient - only about 25% - due to reading frame bias and other intrinsic limitations of the recombination process. That was known of course but the accounting possible in this study is novel. Selection for Mu across the pro-B to pre-B transition is observed, most notably the Vh5-2 (V81X) segment, confirming prior work from the Alt laboratory, but now on the population level. Much of this selection is linked to Jh bias and a presumed disfavored interaction with SLC and the Mu 3rd hypervariable region (HVR).

The first key utility of the population numbers available for this study is the finding of specific 3rd HVR residues with pre-BCR selection (LL 193-238). These findings are novel and implicate a critical role for this site in favored/disfavored interaction with the SLC. One can easily imagine experiments *in vivo* to test the authors' hypotheses.

The second is the authors' observation that certain early/proximal Vh gene segments appear to be major targets for receptor editing events, as inferred by their absence as productive but non-functional Mu rearrangements. The authors' genetic population surveys show that the "missing" fraction of in-frame rearrangements of early/proximal Vh gene segments can be explained plausibly by favored Vh gene replacement. The dynamite point here is that the Vh replacement is riding the pro-B cell of a second, in-frame Mu protein that does not interact with the SLC. Vh replacement acts as a mechanism to ensure single BCR specificity. This is very nice work and again a testable hypothesis (I am not impressed by the *in vitro* culture experiment shown in Fig. 3 given heterogeneity in the actual differentiative stages of the collected pro-B cells).

On this note, we agree with the referee that the subsets in Figure 3C are not completely mechanistically- or marker-defined. The interest here however was in the large mu-negative pro-B population, which is clearly the earliest pro-B population here as it has the lowest frequency (0.28) of VDJs per cell compared to the whole population average of 0.8 (Appendix Table A). We consider this makes the VH Frequency analysis of these cells in Figure 3D highly informative regarding the nature of the earliest VDJs.

To that end, Cowell and Davila references point out the surprising result that conserved cRSS can be found scattered about Vh gene segments, not only in the 3' site associated with the traditional receptor editing observations by Wiegert and others. Obviously, if the physiological role of cRSS in Vh genes is inactivation, their location doesn't really matter.

We thank the referee for this comment and realised our discussion omitted this important point and have added the below paragraph:

Line 815: On this note, the study of Davila et al. reported comparable efficiencies of recombination, using an *in vivo* various other cR. VHR involving these would generate larger, likely non-functional VDJ polypeptides. This raises the possibility that VHR occurring away from the 3' cRSS can also release pro-B cells stalled by a non-pairing μ -chain.

On this note, the study of Davila et al. (Davila et al., 2007) reported comparable efficiencies of recombination, using an *in vitro* assay, between various other cRSSs located upstream and the 3' cRSS. VHR involving these would generate larger, likely non-functional VDJ polypeptides. This raises the possibility that VHR occurring away from the 3' cRSS can also release pro-B cells stalled by a non-pairing μ -chain.

Dear Dr. White,

Thank you for submitting your revised manuscript (EMBOJ-2025-120103R) to The EMBO Journal for our consideration, and for your patience during peer review. Your manuscript has now been seen by the three original referees who had previously assessed the initial version of your work, and we have received their comments (included below).

I am very pleased to say that all referees find the revision very satisfactory and their initially raised criticisms and concerns appropriately addressed. They now all recommend publication of the manuscript in The EMBO Journal with a few remaining requests only for minor corrections (ref. #1) and another round of textual editing (ref. #2). We concur with the referee that the text could be further improved for increased clarity and accessibility, and we kindly request you to submit a final version of your manuscript taking into account the fact that our readership is broad and diverse, consisting of molecular biologists working in all areas of life sciences, and your work and its impact on the community would benefit from increased accessibility of the text.

There are also a few other changes and corrections we need you to make in the final version of your manuscript before we can proceed with its formal acceptance and publication:

- Please provide the e-mail address of the corresponding author on the first (title) page of the revised manuscript.
- Please provide a list of up to 5 relevant keywords after the Abstract of your revised manuscript (preferably broad terms that would enhance online search engine discoverability of the article).
- Please make sure that the full specific URL for the deposited sequencing data is included in the Data availability statement of the manuscript.
- Please rename heading "Competing Interests" to "Disclosure and competing interests statement".
- All Figure panel callouts should be listed sequentially.
- We noticed that callouts for Fig: 2B are missing.
- There are callouts for "Appendix Table", but no such label exists in the Appendix file.
- The heading on the first (title) page of the Appendix PDF file should be "Appendix for" followed by the manuscript's title and a brief Table of Contents including page numbers for the listed items. The nomenclature throughout the Appendix file should be "Appendix Figure S#" and "Appendix Table S#"; please also update all callouts in the main manuscript file accordingly.
- Please make sure that all relevant fields in the Author Checklist are completed as appropriate (for example, no information is currently provided in the "Experimental study design and statistics" section). Please also name the sections of the manuscript where the information is available for each field where a positive response has been indicated (a few are currently missing).
- Thank you for providing your source data, we think that they are a valuable resource for the community. We kindly request you to also include in your source data the numerical data used to generate the plots in your manuscript, according to the following instructions: source data files need to be saved in a scheme one figure/folder and then uploaded as .zip files. E.g. all the Source data files for Figure 1 need to be saved in a single folder and this needs to be zipped and then uploaded as "SD Figure 1.zip" file. For EV and/or Appendix figures, please ZIP together all source data. The completed source data checklist should remain uploaded as Related Manuscript File.
- Please note that EMBO press papers are accompanied online by:
 - A) a short (2 sentences) summary of the findings and their significance,
 - B) 2-5 short bullet points highlighting the key results, and
 - C) a synopsis image in .jpg or .png format that is exactly 550 pixels wide and 300-600 pixels high (the height is variable). Please note that all text needs to be legible at the final size.Please upload this information along with your revised manuscript (the text for A and B should be provided in a separate Word file).
- During our routine data checks, our data editors have raised the following query regarding figure legends: Information related to "n" is missing in the legends of Figures figures 5A, B, D .
- The order of the manuscript sections must be corrected as follows: Title page - Abstract and Keywords - Introduction - Results - Discussion - Methods - Data Availability - Acknowledgements - Disclosure and Competing Interests Statement - References - Figure Legends - main Tables (if there are any) - Expanded View Figure Legends.

Please also note that as part of the EMBO publications' Transparent Editorial Process, The EMBO Journal publishes online a Peer Review File along with each accepted manuscript. This File will be published in conjunction with your paper and will include the referee reports, your point-by-point response and all pertinent correspondence relating to the manuscript. You can opt out of this by letting the editorial office know (contact@embojournal.org). If you do opt out, the Peer Review File link will point to the following statement: "No Peer Review File is available with this article, as the authors have chosen not to make the review process public in this case."

We look forward to seeing a final version of your manuscript as soon as possible. Please let us know if you have any questions and use this link to submit your revision: <https://emboj.msubmit.net/cgi-bin/main.plex>.

Best regards,

Ioannis

Referee #1:

The revised manuscript addresses my concerns appropriately and is significantly improved. It represents an important contribution and I support publication in EMBO J.

I found one line in need of correction:

Line 115: I believe "Basle" should be "Basel" (twice).

Referee #2:

In their revision the authors have addressed my criticisms. The paper contains valuable and extensive new data on a particular mechanism of somatic antibody diversification, namely that V gene replacement. This mechanism had been discovered and studied before by others, but the present study goes beyond previous ones in terms of breadth and defining the impact of V gene replacement along the various phases of B cell development, with an emphasis on the presumed role of non-pairing surrogate light chains in this process. Unfortunately the manuscript remains difficult to read and would profit from another round of stylistic editing. (As an example, here is a section for re-writing:

Lines 122-124: "Whilst this method may be limited by a modest sequence sampling rate of 5-10% of sequences in the starting gDNA pool, (Chovanec et al., 2018), this was overcome with use of bone-marrow from larger pools of mice.")

Referee #3:

The authors' revisions make the MS entirely suitable for publication in EMBO J.

All editorial and formatting issues were resolved by the authors.

Dear Harry,

Congratulations on an excellent manuscript! I am very pleased to inform you that it has been accepted for publication in The EMBO Journal. Thank you for comprehensively addressing the initially raised referees' concerns and all editorial requests for changes and corrections.

If you have any questions, please do not hesitate to contact the Editorial Office. Thank you for your contribution to The EMBO Journal. Working with you has been a pleasure.

Best regards,

Ioannis
